# NEMOTRON-CC-MATH: A 133 BILLION-TOKEN-SCALE HIGH QUALITY MATH PRETRAINING DATASET

**Rabeeh Karimi Mahabadi**[1][*], **Sanjeev Satheesh**[1][*], **Shrimai Prabhumoye**[1,2], **Mostofa Patwary**[1],
**Mohammad Shoeybi**[1], **Bryan Catanzaro**[1]
[1]NVIDIA  [2]Boston University
rkarimimahab@nvidia.com, sasatheesh@nvidia.com

## ABSTRACT

Pretraining large language models (LLMs) on high-quality, structured data such as mathematics and code substantially enhances reasoning capabilities. However, existing math-focused datasets built from Common Crawl suffer from degraded quality due to brittle extraction heuristics, lossy HTML-to-text conversion, and the failure to reliably preserve mathematical structure. In this work, we introduce Nemotron-CC-Math, a large-scale, high-quality mathematical corpus constructed from Common Crawl using a novel, domain-agnostic pipeline specifically designed for robust scientific text extraction.

Unlike previous efforts, our pipeline recovers math across various formats (e.g., MathJax, KaTeX, MathML) by leveraging layout-aware rendering with lynx and a targeted LLM-based cleaning stage. This approach preserves the structural integrity of equations and code blocks while removing boilerplate, standardizing notation into LaTeX representation, and correcting inconsistencies.

We collected a large, high-quality math corpus, namely Nemotron-CC-Math-3+ (133B tokens) and Nemotron-CC-Math-4+ (52B tokens). Notably, Nemotron-CC-Math-4+ not only surpasses all prior open math datasets-including Mega-Math, FineMath, and OpenWebMath-but also contains $5.5\times$ more tokens than FineMath-4+, which was previously the highest-quality math pretraining dataset. When used to pretrain a Nemotron-T 8B model, our corpus yields +4.8 to +12.6 gains on MATH and +4.6 to +14.3 gains on MBPP+ over strong baselines, while also improving general-domain performance on MMLU and MMLU-Stem.

We present the first pipeline to reliably extract scientific content-including math-from noisy web-scale data, yielding measurable gains in math, code, and general reasoning, and setting a new state of the art among open math pretraining corpora. To support open-source efforts, we have released our code[1] and datasets[2].

## 1 INTRODUCTION

The rapid advancement of large language models (LLMs) has sparked a growing interest in improving their mathematical reasoning capabilities. Recent studies indicate that pretraining on carefully curated domain-specialized data—such as mathematics (Paster et al., 2024; Han et al., 2024; Wang et al., 2024b; Azerbayev et al., 2024) and code (Kocetkov et al., 2022; Lozhkov et al., 2024; Li et al., 2023)—substantially improves domain-specific accuracy, general knowledge and reasoning abilities (Muennighoff et al., 2023; Aryabumi et al., 2024; Lewkowycz et al., 2022; Shao et al., 2024). This suggests that high-quality mathematical data plays a pivotal role not only in solving math problems but also in strengthening broader reasoning skills.

Math capabilities of models like O1 (OpenAI, 2024) and DeepSeek-R1 (Guo et al., 2025) critically depend on access to large-scale, high-quality mathematical pretraining data. Unfortunately, datasets used in pretraining SOTA models like DeepSeekMath (Shao et al., 2024), Minerva (Lewkowycz

---

[*]Rabeeh and Sanjeev are the primary authors and contributed equally.
[1]https://github.com/NVIDIA-NeMo/Curator/tree/main/tutorials/math
[2]https://huggingface.co/datasets/nvidia/Nemotron-CC-Math-v1

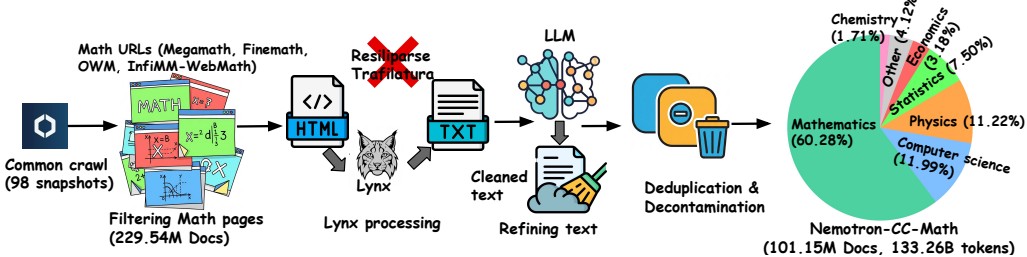

Figure 1: Overview of the Nemotron-CC-Math construction pipeline. Starting from Common Crawl snapshots, we extract math-related URLs using curated datasets (e.g., MegaMath, FineMath). After fetching 229.54M webpages, we render pages through `Lynx` and apply LLM-based cleaning, quality filtering, and deduplication (see §2.1). We visualize the topic distribution of our data (Right).

et al., 2022) and Qwen-2.5-Math (Yang et al., 2024) are not publicly released. Meanwhile, open-source alternatives such as OpenWebMath (OWM) (Paster et al., 2024), FineMath (Allal et al., 2025), InfiMMWebMath (Han et al., 2024) and MathPile (Wang et al., 2024b) remain limited in both scale and fidelity—largely due to brittle extraction pipelines that degrade content quality and fail to preserve mathematical equations and structure (see Appendix A.3).

While Common Crawl forms a primary source for large-scale pretraining (Penedo et al., 2023; Tang et al., 2024; Su et al., 2025), its value for mathematical pretraining remains underexploited. Existing math-specific extraction pipelines (Paster et al., 2024; Zhou et al., 2025) are not well-suited to fully leverage this resource. In particular, current methods frequently fail to detect or accurately extract equations, either omitting them altogether or corrupting their structure (Han et al., 2024; Allal et al., 2025). This severely compromises content fidelity. Mathematical notation on the web appears in a wide range of formats—including MathML, LaTeX, and dynamically rendered scripts—whose representations continue to evolve over time (see Figure 2). Compounding this challenge, HTML pages in Common Crawl often lack associated stylesheets and JavaScript resources, preventing proper rendering and further obstructing reliable equation recovery. These limitations collectively hinder the construction of high-quality mathematical pretraining corpora that capture the breadth and variety of real-world mathematical content.

To bridge this gap, we propose a modular, scalable, and domain-agnostic framework for reliably extracting mathematically rich content from raw web data, enabling the construction of a large-scale, diverse, and high-fidelity math corpus. Our multi-stage extraction and filtering pipeline ensures quality at scale (see Figure 1). In the first stage, HTML documents are rendered into text using the `Lynx` text-based browser[3], which preserves mathematical equations and symbols with high accuracy. In the second stage, a lightweight LLM normalizes heterogeneous math representations into LaTeX while discarding boilerplate and irrelevant content. This LLM-based approach allows us to avoid the brittle, heuristic-based rules employed in previous pipelines (Paster et al., 2024), resulting in more reliable and consistent extraction of mathematical content. Subsequently, we apply a quality classifier to retain the high-quality pages, followed by deduplication to eliminate redundancy. In addition, we perform thorough contamination detection against downstream benchmarks (see § 2.4), ensuring that any overlapping or duplicated samples are identified and removed from the corpus.

By leveraging the scale of Common Crawl and the rigor of our pipeline, we present Nemotron-CC-Math—the highest quality open-source math corpus to date, comprising of 133B tokens, where its highest quality subset (Nemotron-CC-Math-4+) totals 53B tokens. Our pipeline is optimized for performance using Polars and Ray, enabling us to process terabytes of HTML content efficiently. To facilitate future research, we release both the dataset and our full pipeline implementation.

Our contributions are as follows:

- We reviewed prior data extraction pipelines, and show that they fail to accurately extract math and code content, often stripping math equations and code snippets (Appendix A.3).
- We introduce a scalable and modular pipeline for extracting high-quality mathematical content from Common Crawl, explicitly addressing the longstanding challenge of HTML math variability-including LaTeX, MathML, Unicode, and inline or malformed equations.

---

[3] https://lynx.invisible-island.net/

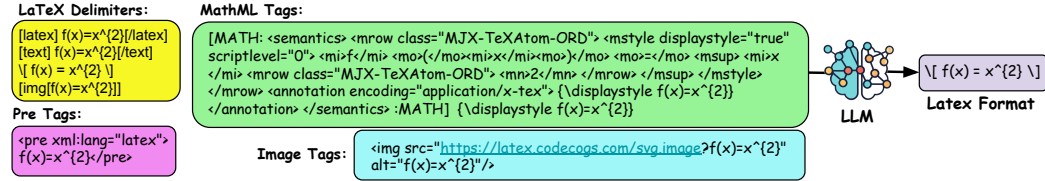

Figure 2: Mathematical expressions on HTML pages appear in diverse formats—LaTeX within custom delimiters, `<pre>` blocks, image tags, and MathML. These variations challenge standard text extraction pipelines, which often fail to recover the underlying LaTeX equations correctly. To address this, we use an LLM to standardize all mathematical representations into a unified LaTeX format.

- To our knowledge, this is the first work to employ the text-based browser `Lynx` for HTML-to-plain-text conversion with preservation of math and code formatting, and to introduce LLM-based standardization of mathematical representations across the content.

- We will release Nemotron-CC-Math, a dataset of 133B tokens of high-quality and diverse math-rich web documents extracted from Common Crawl, whose 4+ subset contains 5.5× more tokens than FineMath-4+, the previous best math pretraining dataset.

- We will open-source our full pipeline (extraction, processing and scoring) to ensure reproducibility and support its application to other domains.

- We thoroughly analyze Nemotron-CC-Math by examining its composition, including statistics on webpage types, subject areas, and the most frequent source domains.

- Through extensive experiments and detailed quality analysis (§3.3), we demonstrate that models pretrained on our dataset outperform those trained on existing pretraining math datasets across a range of benchmarks, including math, code, and general knowledge tasks.

## 2 THE COLLECTION OF NEMOTRON-CC-MATH

We construct the Nemotron-CC-Math corpus from Common Crawl[4], a large-scale web archive extensively used in recent LLM training (Dubey et al., 2024; Hui et al., 2024; DeepSeek-AI et al., 2025). Common Crawl contains over 300B documents across more than 6M WARC files (each contains over 1GB of compressed content). Our goal is to build a pipeline that can process the technical content from Common Crawl correctly. We apply our pipeline to math domain to assemble a high-quality, large-scale corpus of mathematical content from Common Crawl. To achieve this, we designed a robust and highly scalable data processing pipeline capable of operating at the full scale of Common Crawl, as illustrated in Figure 1.

Prior efforts such as OWM (Paster et al., 2024) and DeepSeekMath (Shao et al., 2024) rely on lightweight classifiers to identify technical pages. We initially explored a similar approach but found fundamental limitations in achieving meaningful improvements: first, mathematical content constitutes $< 1\%$ of Common Crawl, making manual ground truth annotation extremely difficult; second, since classifiers must run on all Common Crawl documents, only very efficient methods like FastText with simplified HTML parsing are viable. This creates an inherently high-bias setup with no straightforward path to improvement—attempts to increase recall for technical content invariably lead to drastic drops in precision. Rather than refining such classifiers for marginal gains, we leverage community-filtered datasets: extracting URLs from OWM, InfiMM-WebMath (Han et al., 2024), FineMath (Allal et al., 2025), and MegaMath (Zhou et al., 2025), including all major subsets. This approach allows us to benefit from the diverse filtering strategies employed by different research groups while avoiding the limitations of any single classifier.

We then retrieve the original HTML from 98 Common Crawl snapshots (2014-2024) for these URLs, enabling fine-grained extraction that preserves mathematical expressions, symbols, and formatting-often degraded in prior processing (Appendix A.3). This process yields 229.54M high-quality webpages spanning a diverse range of mathematical content.

---

[4]https://commoncrawl.org/

## 2.1 RELIABLE TEXT EXTRACTION FOR SCIENTIFIC CONTENT

### 2.1.1 LIMITATIONS OF PRIOR WORK

Extracting mathematical content from raw HTML presents a significant challenge for text extraction pipelines. Unlike natural language, which often follows consistent structural patterns, math equations appear in highly variable forms across the web (see Figure 2). These variations stem from the absence of standardized conventions for embedding math in HTML, as well as the diversity of rendering engines (e.g., MathJax, KaTeX, MathML, image-based representations, and custom plugins). Moreover, websites frequently evolve their rendering strategies, making any fixed set of heuristics fragile in practice. As a result, existing extraction pipelines often fail to reliably extract scientific content, with equations either missed entirely, mis-parsed, or distorted. Preserving formatting is equally important: the indentation and layout of code blocks and the placement of mathematical symbols often carry semantic meaning, and losing this structure severely degrades the value of the extracted content for downstream modeling.

Existing content extraction tools such as JUSTEXT (Endrédy & Novák, 2013), TRAFILATURA (Barbaresi, 2021), and RESILIPARSE (Bevendorff et al., 2018)—used in large-scale dataset construction pipelines including The Pile (Gao et al., 2020), FineMath (Allal et al., 2025), and RefinedWeb (Penedo et al., 2023)—were designed primarily for general-purpose boilerplate removal and narrative text extraction. While effective for general documents, they often strip or corrupt equations, miss inline LaTeX equations changing semantics, and flatten (or miss) code blocks requiring strict indentation (e.g., Python). These shortcomings limit their usability for building high-quality math or code datasets. Examples are provided in Appendix A.3.

### 2.1.2 OUR TEXT EXTRACTION PIPELINE

The diversity of mathematical representations on the web necessitates using large language models to faithfully convert technical HTML content into a format suitable for LLM pretraining. Since raw HTML from WARC files is too verbose for direct LLM processing, and traditional parsers risk losing critical information, we employ `lynx`, a text-based browser that renders web pages into plain text while preserving mathematical equations and code formatting. Unlike DOM-based parsers used in prior work (Paster et al., 2024; Zhou et al., 2025; Allal et al., 2025), `lynx` executes HTML layout rules to produce output that mirrors the human-perceived page structure, reliably capturing equations and maintaining code indentation.

While `lynx` preserves the structural layout of web pages, its output includes boilerplate elements such as navigation bars and redundant headers. To refine this output, we apply an LLM cleanup pass using Phi-4 (Abdin et al., 2024a)(14B parameters), which preserves primary content and references while removing non-essential content. LLM additionally standardizes mathematical expressions into consistent LaTeX format (Figure 2), and corrects typographical errors. This two-stage pipeline-structural preservation via `lynx` followed by semantic refinement via an LLM-yields high-quality, coherent text suitable for large-scale mathematical corpora. Ablation studies (§3.2) show that this cleanup task is simple enough for smaller models to perform effectively. Qualitative comparisons with prior work and the full cleanup prompt are provided in Appendices A.3 and A.5, respectively.

## 2.2 QUALITY CLASSIFICATION

To support the later stages of training where data fidelity is especially important (Hu et al., 2024; Abdin et al., 2024b), we further filtered our Nemotron-CC-Math to retain only its highest-scoring subset, Nemotron-CC-Math-4+. We employed the FineMath classifier (Allal et al., 2025) which assigns a 5-point scale score to each page, focusing on identifying content with mathematical reasoning and material suited to middle- and high-school levels. The data distribution for each quality score is provided in Appendix §A.7. After classification, we also performed deduplicatication and decontamination (see §2.3 and §2.4). We developed two variants of Nemotron-CC-Math: Nemotron-CC-Math-4+ (52.32B tokens, 45M documents) with scores 4-5 and Nemotron-CC-Math-3+ (133.26B tokens, 101M documents) with scores 3-5.

Table 1: Comparison of Nemotron-CC-Math with math pretraining datasets. Nemotron-CC-Math-4+ is 5.5× larger than the highest-quality open math dataset (FineMath-4+) with a permissive license, and substantially outperforms FineMath across math, code, and knowledge tasks (Table 2).

| Dataset | Open Source | #Documents (M) | #Tokens (B) | Source |
|---|---|---|---|---|
| Minerva (Lewkowycz et al., 2022) | ✗ | - | 38.50 | arXiv, Web |
| MathMix (Lightman et al., 2023) | ✗ | - | 1.50 | Unknown |
| DeepSeekMath (Shao et al., 2024) | ✗ | - | 120 | CommonCrawl |
| ProofPile (Azerbayev et al., 2023) | ✓ | 2.04 | 8.30 | arXiv, Textbooks, Formal Math Libraries, StackExchange, ProofWiki, MATH |
| ProofPile-2 (Azerbayev et al., 2024) | ✓ | 11.20 | 55 | OpenWebMath, ArXiv, AlgebraicStack |
| AMPS (Hendrycks et al., 2021b) | ✓ | 5.10 | 0.70 | Khan Academy, Synthetic data |
| MathPile (Wang et al., 2024b) | ✓ | 0.73 | 9.50 | arXiv, Textbooks, ProofWiki, Wikipedia, StackExchange, CommonCrawl |
| OpenWebMath (Paster et al., 2024) | ✓ | 6.30 | 14.70 | CommonCrawl |
| InfiMM-WebMath-4+ (Han et al., 2024) | ✓ | 6.30 | 8.50 | CommonCrawl |
| FineMath-4+ (Allal et al., 2025) | ✓ | 6.70 | 9.60 | CommonCrawl |
| MegaMath-Pro (Zhou et al., 2025) | ✓ | 15 | 15.10 | CommonCrawl |
| Nemotron-CC-Math-4+ (Ours) | ✓ | 45.10 | 52.32 | CommonCrawl |
| InfiMM-WebMath-3+ (Han et al., 2024) | ✓ | 13.90 | 20.50 | CommonCrawl |
| FineMath-3+ (Allal et al., 2025) | ✓ | 21.40 | 34 | CommonCrawl |
| MegaMath-Web (Zhou et al., 2025) | ✓ | 106.50 | 263.90 | CommonCrawl |
| Nemotron-CC-Math-3+ (Ours) | ✓ | 101.15 | 133.26 | CommonCrawl |

## 2.3 FUZZY DEDUPLICATION

Removing near-duplicate documents is essential for efficient and stable model training, and reducing the risk of memorization (Lee et al., 2022; Tokpanov et al., 2024). We applied fuzzy deduplication using the NeMo-Curator framework, which uses a MinHash-based Locality Sensitive Hashing (LSH) (Broder, 2000) to efficiently detect duplicates. The probability that two documents with Jaccard similarity $S$ hash to the same bucket is $P = 1 - (1 - S^b)^r$, where $b$ is the number of hash functions per band and $r$ is the number of bands. With $r=20$ bands and $b=13$ hash functions per band, our setup targets a Jaccard similarity threshold of 0.8. Pairwise similarity is computed using 24-character n-grams, and LSH uses concurrent shuffling of five bands to identify duplicates.

## 2.4 DECONTAMINATION

The source documents used in Nemotron-CC-Math are from mostly pre-decontaminated datasets. However, we follow a more thorough decontamination procedure as outlined in Yang et al. (2023). We embed all the documents in Nemotron-CC-Math using the Qwen2.5B 32B model (Qwen et al., 2025) as well as all the prompts and answers from our evaluation benchmarks: MMLU (Hendrycks et al., 2021a), MMLU-Pro (Wang et al., 2024a), MATH (Hendrycks et al., 2021b), and GSM8K (Cobbe et al., 2021). We remove all documents with a cosine similarity above 0.9 to any benchmark prompt or answer, resulting in the removal of less than 0.002% of all documents.

## 3 EXPERIMENTS

**Datasets** We compare Nemotron-CC-Math to existing prior math pretraining datasets, including Megamath, OWM, and FineMath. Table 1 summarizes the dataset statistics.

**Experimental Setup** Math and code abilities generally arise only after extensive training; following Blakeney et al. (2024); Dubey et al. (2024); OLMo et al. (2024); Allal et al. (2025) to estimate the quality of different math pretraining datasets, we run annealing ablations on a mid training checkpoint of Nemotron-T 8B model (NVIDIA et al., 2025). The base model was pretrained on 9T tokens using a mixture of general-domain and math-focused corpora (see Appendix A.8 for a detailed breakdown). In each ablation, the target math dataset is upweighted to constitute 30% of the total data blend, while the weights of all other datasets are correspondingly downweighted to make up the remaining 70%. This controlled adjustment isolates the contribution of the math data while preserving overall blend composition (See Appendix A.4 for hyper-parameters). We consider two controlled ablations:

- **100B Token Ablation:** This setting targets compact, high-quality math datasets, typically below 30B tokens. For each run, the mathematical portion of the blend is replaced with a single candidate dataset-such as FineMath-4+, MegaMath-Pro, or OWM—enabling direct comparison with Nemotron-CC-Math-4+. The modified blends are trained for 100B tokens to evaluate performance under a consistent compute budget.

- **300B Token Ablation:** To fairly assess larger math datasets, including FineMath-3+ and MegaMath-Web, we apply the same replacement and proportional adjustment strategy but extend the total annealing budget to 300B tokens. This configuration also tests whether increased scale can offset dataset quality differences.

**Benchmarks** We evaluate model performance across a diverse suite of benchmarks spanning knowledge understanding, code, and mathematical reasoning tasks. Knowledge understanding is assessed using MMLU datasets, including MMLU-Pro (Wang et al., 2024a), MMLU, and MMLU-STEM (Hendrycks et al., 2021a) with results reported as exact match (EM) accuracy. Code generation quality is measured on four tasks-MBPP (Austin et al., 2021), and HumanEval (Chen et al., 2021) and their EvalPlus variants, HumanEval+ and MBPP+ (Liu et al., 2023). For code tasks, following Guo et al. (2025), to improve the stability, we report the $\text{avg}@20$ which reports the average accuracy from generating 20 samples for each prompt. To produce these samples, we apply nucleus sampling with a temperature of 0.6 and a top-$p$ value of 0.95. Mathematical reasoning is evaluated on the GMS8K (Cobbe et al., 2021) and MATH (Hendrycks et al., 2021b) benchmarks, with greedy decoding and using Math-Verify[5] for symbolic matching. We run evaluations of all models ourselves using `lm-evaluation-harness`[6].

## 3.1 Pretraining Experiments Results

**Results** Tables 2 compare Nemotron-T 8B models pretrained with different math datasets at 100B and 300B tokens, respectively. Across all benchmarks, models using the curated Nemotron-CC-Math consistently match or outperform competing datasets—including OWM, MegaMath, and FineMath—on knowledge, code, and math tasks.

At 100B tokens, Nemotron-CC-Math-4+ achieves the top results in every math task-e.g., 40.6 on MATH (+4.8 vs. FineMath-4+, +6.6 vs. MegaMath-Pro) and 76.27 on GMS8K (+0.3 vs. FineMath-4+, +4.85 vs. OWM). It also leads most code benchmarks (e.g., 34.82 on HumanEval+, +2.3 vs. OWM) and all knowledge tasks (e.g., 38.49 on MMLU-Pro, +2.1 vs. MegaMath-Pro).

At 300B tokens, Nemotron-CC-Math-3+ extends these gains-reaching 44.2 on MATH (+9.6 vs. FineMath-3+, +12.6 vs. MegaMath-Web) and 80.06 on GMS8K (+0.6 vs. FineMath-3+, +3.6 vs. OWM). Code scores also improve substantially, with 37.16 on HumanEval+ (+3.0 vs. FineMath-3+) and 43.51 on MBPP+ (+4.6 vs. MegaMath-Web, +14.32 vs. Finemath-3+). Knowledge remains best or near-best across MMLU variants, with a top score of 64.26 on MMLU-STEM.

Although we do not explicitly target the code domain, it is noteworthy that the curated Nemotron-CC-Math datasets substantially improve code performance. Upon analysis, we find that Nemotron-CC-Math-3+ and Nemotron-CC-Math-4+ contain approximately 4.3M and 1.44M samples with code snippets[7]. In contrast to prior datasets, which often fail to capture code content, our curation pipeline retains code snippets in full, preserving syntax and structure. We attribute the observed code improvements to this incidental yet high-quality code data. Overall, results show that high-quality curated math data in pretraining boosts performance in math reasoning, code, and general knowledge. Comparing 100B and 300B token results, gains scale with more pretraining. This highlights the value of high-quality math data for improving LLMs across specialized and general domains.

## 3.2 Ablation On Model Choice

To ablate the model choice for the task of boilerplate removal from rendered web pages, we sampled 7M documents and evaluated several instruction-tuned LLMs including DeepSeek-V3 (Liu et al.,

---

[5]`https://github.com/huggingface/math-verify`
[6]`https://github.com/EleutherAI/lm-evaluation-harness`.
[7]We filter out examples enclosed within triple backticks indicating a code block (e.g., "'python ... "').

| Models Trained on 100B Tokens | | | | |
|---|---|---|---|---|
| **Benchmark** (Metric) | **OWM** | **MegaMath (Pro)** | **FineMath (4+)** | **Nemotron-CC-Math (4+)** |
| **Knowledge** MMLU-Pro (EM) | 35.49 | 36.41 | 36.74 | **38.49** |
| MMLU (EM) | 65.62 | 66.81 | 66.73 | **67.55** |
| MMLU-Stem (EM) | 58.83 | 60.86 | 61.62 | **62.67** |
| **Code** HumanEval+ (avg@20) | 32.53 | 31.01 | 32.16 | **34.82** |
| MBPP+ (avg@20) | 43.76 | **46.03** | 28.88 | 45.11 |
| MBPP (avg@20) | 53.11 | 52.51 | 53.42 | **53.48** |
| HumanEval (avg@20) | 37.07 | 35.91 | 37.77 | **38.93** |
| **Math** MATH (EM) | 29.20 | 34.00 | 35.80 | **40.60** |
| GMS8K (EM) | 71.42 | 73.46 | 75.97 | **76.27** |
| Models Trained on 300B Tokens | | | | |
| **Benchmark** (Metric) | **OWM** | **MegaMath (Web)** | **FineMath (3+)** | **Nemotron-CC-Math (3+)** |
| **Knowledge** MMLU-Pro (EM) | 35.00 | 36.33 | **39.57** | 39.32 |
| MMLU (EM) | 65.20 | 65.44 | 67.92 | **68.20** |
| MMLU-Stem (EM) | 59.20 | 59.88 | 62.29 | **64.26** |
| **Code** HumanEval+ (avg@20) | 33.54 | 32.29 | 34.18 | **37.16** |
| MBPP+ (avg@20) | 37.59 | 38.89 | 29.19 | **43.51** |
| MBPP (avg@20) | 52.22 | 53.05 | **57.57** | 56.15 |
| HumanEval (avg@20) | 38.32 | 36.34 | 37.80 | **40.30** |
| **Math** MATH (EM) | 34.20 | 31.60 | 34.60 | **44.20** |
| GMS8K (EM) | 76.42 | 78.24 | 79.45 | **80.06** |

Table 2: Evaluation results for models trained with different math datasets using either 100B or 300B tokens. NEMOTRON-CC-MATH variants consistently outperform or obtain comparable results to OpenWebMath, MegaMath, and FineMath baselines across knowledge, code, and math tasks. Math performance improves with a longer token horizon, showing Nemotron-CC-Math continues to scale effectively with increased training. Code results use average accuracy over 20 generations; all other tasks use exact match (EM). Bold indicates the best result in each row.

2024), Qwen2.5-32B/Instruct, Qwen2.5-72B/Instruct (Team, 2024), and Phi-4 across knowledge, coding, and math benchmarks.

Table 3 presents the results. Surprisingly, despite its significantly smaller size (14B parameters), Phi-4 performs competitively across all domains, often matching or exceeding the results of much larger models such as DeepSeek-V3 (671B) and Qwen2.5-72B-Instruct (72B). In particular, Phi-4 achieves the best performance on math tasks (e.g., 79.98 EM on GMS8K and 40.6 EM on MATH) and leads or matches the performance in several code-related benchmarks.

Given the marginal differences in performance and the substantial gap in computational cost, we selected Phi-4 as the default model for all experiments in this paper. Our findings indicate that the task of webpage boilerplate removal does not require excessively large models, and smaller instruction-tuned models can yield efficient and effective results.

### 3.3 LLM-AIDED QUALITY ASSESSMENT OF SCIENTIFIC CONTENT FIDELITY

We first performed an overlap analysis across OWM, MegaMath-Pro, FineMath, and our Nemotron-CC-Math datasets, identifying 97,788 shared samples. As this joint subset is far too small for meaningful side-by-side pretraining experiments, we use it as a rigorous, shared basis for comparative quality assessment. To complement standard benchmark results, we conduct an LLM-aided quality assessment to directly measure how effectively Common-Crawl-derived mathematical datasets preserve scientific content during extraction and cleaning. This assessment focuses on four critical dimensions essential for high-quality mathematical pretraining corpora:

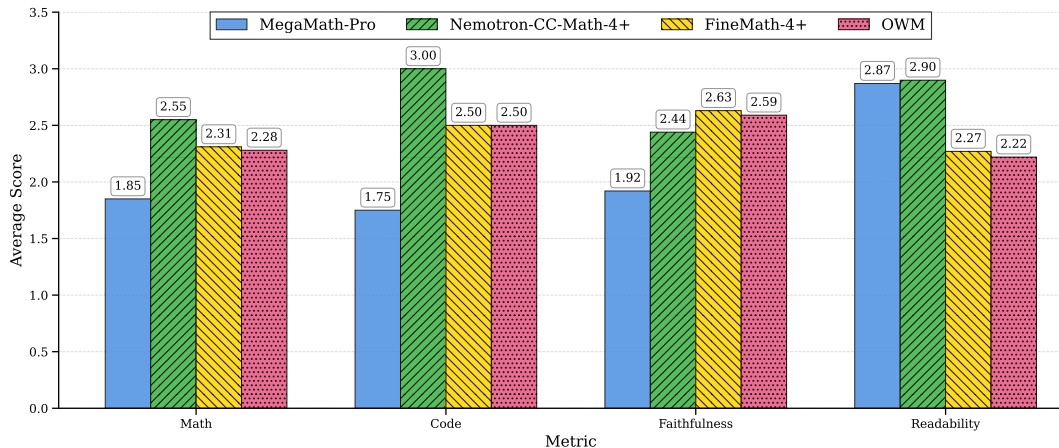

Figure 3: LLM-Aided quality assessment results comparing the cleaned webpage quality across four datasets: Ours (Nemotron-CC-Math), OWM, FineMath, and MegaMath. The LLM judge scored 100 randomly sampled documents from each dataset on four dimensions. Our method achieves the strongest performance in preserving mathematical expressions and code structure, with competitive faithfulness and the highest readability. These results highlight the effectiveness of the Lynx → LLM pipeline in retaining scientific structure while improving textual clarity.

- **Math Preservation (0–3 or N/A):** Correctness and completeness of mathematical expressions unified to LaTeX.

- **Code Preservation (0–3 or N/A):** Structural and semantic integrity, syntax, and functional behavior of preserved code blocks.

- **Faithfulness (0–3):** Preservation of core scientific content integrity without omission or meaning alteration.

- **Readability (0–3):** Overall clarity, organization, coherence, and textual flow of the output document.

**Methodology**   We randomly sampled 100 webpages from the shared subset. We employed OpenAI `gpt-5.1` as an automated judge to assess the conversion quality. The judge was provided with the original raw text (extracted via `Lynx`) and the converted document from each dataset, guided by a detailed scoring rubric and evaluation instructions (see Appendix A.6). We report the mean score for each dataset across the 100 samples and four dimensions.

**Results**   Figure 3 summarizes the assessment results. Nemotron-CC-Math demonstrates superior fidelity in preserving the underlying scientific structure, achieving the highest scores in both Math and Code Preservation:

| | Benchmark (Metric) | DeepSeek-V3 | Qwen2.5-32B | Qwen2.5-72B | Phi-4 |
|---|---|---|---|---|---|
| **Knowledge** | MMLU-Pro (EM) | 38.82 | **39.65** | **39.65** | 38.49 |
| | MMLU (EM) | 67.68 | 67.01 | **67.73** | 67.54 |
| | MMLU-Stem (EM) | 62.96 | 61.88 | 62.73 | **63.24** |
| **Code** | HumanEval+ (avg@20) | 28.54 | 28.35 | 29.63 | **30.40** |
| | MBPP+ (avg@20) | **45.99** | 38.58 | 41.34 | 41.88 |
| | MBPP (avg@20) | 53.91 | 53.83 | 54.09 | **55.39** |
| | HumanEval (avg@20) | 32.10 | 31.92 | **35.21** | 34.09 |
| **Math** | MATH (EM) | 36.60 | 38.80 | 38.60 | **40.60** |
| | GMS8K (EM) | 75.51 | 74.00 | 73.92 | **79.98** |

Table 3: Model choice ablation. We compare DeepSeek-V3 (671B), Qwen2.5-32B/72B, and Phi-4 (14B) across benchmarks. Despite its smaller size, Phi-4 performs competitively—often leading in math—demonstrating smaller models can efficiently clean webpages without losing performance.

| Domain | #Documents (M) | Document % | | Domain | #Characters (B) | Characters % |
|---|---|---|---|---|---|---|
| mathhelpforum.com | 8.54 | 8.44 | | mathhelpforum.com | 17.11 | 3.67 |
| jiskha.com | 5.33 | 5.26 | | jiskha.com | 12.52 | 2.69 |
| physicsforums.com | 2.82 | 2.78 | | mathforum.org | 8.96 | 1.92 |
| math.stackexchange.com | 2.38 | 2.35 | | physicsforums.com | 8.19 | 1.76 |
| mathforum.org | 2.38 | 2.35 | | math.stackexchange.com | 6.96 | 1.49 |
| openstudy.com | 1.88 | 1.86 | | mathoverflow.net | 6.78 | 1.45 |
| forums.wolfram.com | 1.51 | 1.49 | | nrich.maths.org | 6.24 | 1.34 |
| mathoverflow.net | 1.33 | 1.32 | | scribd.com | 4.47 | 0.96 |
| nrich.maths.org | 1.13 | 1.12 | | educator.com | 3.58 | 0.77 |
| mathisfunforum.com | 0.76 | 0.75 | | forums.wolfram.com | 3.35 | 0.72 |
| coursehero.com | 0.76 | 0.75 | | docplayer.net | 3.20 | 0.69 |
| brilliant.org | 0.68 | 0.67 | | en.wikipedia.org | 3.07 | 0.66 |
| gmatclub.com | 0.65 | 0.65 | | openstudy.com | 3.10 | 0.66 |
| chegg.com | 0.58 | 0.57 | | gmatclub.com | 2.73 | 0.59 |
| gradesaver.com | 0.54 | 0.53 | | mathisfunforum.com | 2.72 | 0.58 |
| socratic.org | 0.49 | 0.49 | | coursehero.com | 2.31 | 0.50 |
| purplemath.com | 0.45 | 0.44 | | slideplayer.com | 2.06 | 0.44 |
| physics.stackexchange.com | 0.44 | 0.43 | | hindawi.com | 1.98 | 0.42 |
| betterlesson.com | 0.41 | 0.41 | | softmath.com | 1.95 | 0.42 |
| brainmass.com | 0.40 | 0.40 | | archive.org | 1.96 | 0.42 |

Table 4: Comparison of Most Common Domains by Document (left) and Character Count (right).

- **Math Preservation (2.55):** Our score significantly surpasses all baselines (OWM: 2.28, FineMath: 2.31, MegaMath: 1.85). This validates that our combined `lynx`-based extraction and LLM-driven LaTeX normalization more reliably preserves mathematical content than prior heuristic HTML-to-text pipelines.

- **Code Preservation (3.00):** Nemotron-CC-Math achieved the maximum score, substantially outperforming all competitors (OWM/FineMath: 2.50, MegaMath: 1.75). This confirms the effectiveness of the `lynx`-based approach in retaining code formatting and indentation, often lost by DOM-based extractors.

- **Faithfulness (2.44) and Readability (2.90):** Although our faithfulness score is marginally lower than FineMath (2.63) and OWM (2.59)—a consequence of the intentional LLM-based cleanup that may involve compressing or rewriting contextual details—Nemotron-CC-Math achieved the highest readability. This demonstrates that our pipeline successfully trades a minimal loss in literal text preservation for superior textual coherence and clarity.

In conclusion, these results confirm that Nemotron-CC-Math offers the optimal balance between structural fidelity and textual clarity. By moving beyond brittle heuristic pipelines, our Lynx → LLM approach captures essential scientific content and structure while yielding coherent, high-quality text, establishing Nemotron-CC-Math as a substantially higher-quality scientific corpus compared to existing Common-Crawl-based math pretrainig datasets.

## 3.4 DATASET ANALYSIS

**Data Composition** We measured domain distribution by document and character count. Table 4 shows the top twenty domains by each metric. Similar to prior work (Paster et al., 2024), the most common sources are discussion forums, Q&A sites, and educational resources. Overall, the dataset spans 980,922 unique domains, with the top 100 domains accounting for 36.46% of characters and 43.22% of documents.

**Topic Distribution** To characterize the dataset, we randomly sampled 150,000 documents and classified them into mathematics, physics, statistics, chemistry, economics, computer science, or other using the Qwen3-30B-A3B-Instruct-2507 model (see Appendix A.2 for the prompt). Figure 1 shows the results. Most documents pertain to mathematics, with the remainder distributed across the other scientific domains; approximately 4.12% fall outside these categories.

## 4 RELATED WORKS

High-quality math pretraining datasets are essential for improving LLM reasoning. OWM compiles 14.7B tokens from Common Crawl but relies on brittle heuristics and Resiliparse for HTML rendering, often stripping or corrupting formulas and code. FineMath inherits these issues, building its 54B-token corpus using the OWM pipeline. Similarly, MegaMath faces similar challenges.

MathPile (Wang et al., 2024b) aggregates 9.5B tokens from sources including arXiv (85%), textbooks, and forums but much of the content remains in raw LaTeX form, limiting usability for LLM pretraining. InfiMM-Web-Math (Han et al., 2024) is 40B tokens multimodal dataset pairing images with math text. Proof-Pile (Azerbayev et al., 2023) is a 8.3B-token dataset collected from various sources such as arXiv, formal math libraries, Wikipedia and Stack Exchange. Proof-Pile-2 (Azerbayev et al., 2024) is a 55B-token dataset combining arXiv, OWM, and Algebraic-Stack mathematical code. Additionally, auxiliary datasets include AMPS (Hendrycks et al., 2021b) with Khan Academy problems and Mathematica-generated content, and NaturalProofs (Welleck et al., 2021), covering theorems and proofs from ProofWiki and the Stacks project.

Proprietary datasets like WebMath (OpenAI) (Polu & Sutskever, 2020), MathMix (Lightman et al., 2023), DeepSeekMath (Shao et al., 2024), and Minerva's Math Web Pages (Lewkowycz et al., 2022) advance math reasoning but lack public access, limiting transparency. We release Nemotron-CC-Math openly to foster community progress.

## 5 CONCLUSION

We present a scalable, domain-agnostic pipeline for extracting high-quality technical content from Common Crawl, focusing here on the mathematical domain. By integrating robust HTML-to-text conversion with LLM-based domain-aware cleaning, our approach addresses longstanding challenges in web-scale extraction of structured technical data.

Applied to the math domain, our pipeline produced Nemotron-CC-Math, whose 4+ subset is $5.5\times$ larger than the previous highest-quality math set, FineMath-4+. Pretraining on Nemotron-CC-Math improves math reasoning (+12.6 MATH), code generation (+14.3 MBPP+), and general knowledge (+5.1 MMLU-Stem), outperforming prior math datasets.

Importantly, the modular, domain-agnostic design enables application to other technical fields. As LLMs advance, our pipeline offers a crucial tool for generating targeted, high-quality pretraining data to drive model capabilities.

ACKNOWLEDGMENTS

We would like to thank Markus Kliegl for the original suggestion to explore text based browsers, and Ying Lin for their helpful comments and suggestions.

REFERENCES

Marah Abdin, Jyoti Aneja, Harkirat Behl, Sébastien Bubeck, Ronen Eldan, Suriya Gunasekar, Michael Harrison, Russell J Hewett, Mojan Javaheripi, Piero Kauffmann, et al. Phi-4 technical report. *arXiv preprint arXiv:2412.08905*, 2024a.

Marah I Abdin, Sam Ade Jacobs, Ammar Ahmad Awan, Jyoti Aneja, Ahmed Awadallah, Hany Awadalla, Nguyen Bach, Amit Bahree, Arash Bakhtiari, Harkirat S. Behl, Alon Benhaim, Misha Bilenko, Johan Bjorck, Sébastien Bubeck, Martin Cai, Caio César Teodoro Mendes, Weizhu Chen, Vishrav Chaudhary, Parul Chopra, Allie Del Giorno, Gustavo de Rosa, Matthew Dixon, Ronen Eldan, Dan Iter, Amit Garg, Abhishek Goswami, Suriya Gunasekar, Emman Haider, Junheng Hao, Russell J. Hewett, Jamie Huynh, Mojan Javaheripi, Xin Jin, Piero Kauffmann, Nikos Karampatziakis, Dongwoo Kim, Mahoud Khademi, Lev Kurilenko, James R. Lee, Yin Tat Lee, Yuanzhi Li, Chen Liang, Weishung Liu, Eric Lin, Zeqi Lin, Piyush Madan, Arindam Mitra, Hardik Modi, Anh Nguyen, Brandon Norick, Barun Patra, Daniel Perez-Becker, Thomas Portet, Reid Pryzant, Heyang Qin, Marko Radmilac, Corby Rosset, Sambudha Roy, Olatunji Ruwase, Olli Saarikivi, Amin Saied, Adil Salim, Michael Santacroce, Shital Shah, Ning Shang, Hiteshi Sharma, Xia Song, Masahiro Tanaka, Xin Wang, Rachel Ward, Guanhua Wang, Philipp Witte, Michael Wyatt, Can Xu, Jiahang Xu, Sonali Yadav, Fan Yang, Ziyi Yang, Donghan Yu, Chengruidong Zhang, Cyril Zhang, Jianwen Zhang, Li Lyna Zhang, Yi Zhang, Yue Zhang, Yunan Zhang, and Xiren Zhou. Phi-3 technical report: A highly capable language model locally on your phone. *CoRR*, abs/2404.14219, 2024b. URL https://doi.org/10.48550/arXiv.2404.14219.

Loubna Ben Allal, Anton Lozhkov, Elie Bakouch, Gabriel Martín Blázquez, Guilherme Penedo, Lewis Tunstall, Andrés Marafioti, Hynek Kydlíček, Agustín Piqueres Lajarín, Vaibhav Srivastav,

Joshua Lochner, Caleb Fahlgren, Xuan-Son Nguyen, Clémentine Fourrier, Ben Burtenshaw, Hugo Larcher, Haojun Zhao, Cyril Zakka, Mathieu Morlon, Colin Raffel, Leandro von Werra, and Thomas Wolf. Smollm2: When smol goes big – data-centric training of a small language model, 2025. URL `https://arxiv.org/abs/2502.02737`.

Viraat Aryabumi, Yixuan Su, Raymond Ma, Adrien Morisot, Ivan Zhang, Acyr Locatelli, Marzieh Fadaee, Ahmet Üstün, and Sara Hooker. To code or not to code? exploring impact of code in pre-training. In *ICLR*, 2024.

Jacob Austin, Augustus Odena, Maxwell Nye, Maarten Bosma, Henryk Michalewski, David Dohan, Ellen Jiang, Carrie Cai, Michael Terry, Quoc Le, and Charles Sutton. Program Synthesis with Large Language Models, 2021. URL `https://arxiv.org/abs/2108.07732`.

Zhangir Azerbayev, Bartosz Piotrowski, Hailey Schoelkopf, Edward W Ayers, Dragomir Radev, and Jeremy Avigad. Proofnet: Autoformalizing and formally proving undergraduate-level mathematics. *arXiv preprint arXiv:2302.12433*, 2023.

Zhangir Azerbayev, Hailey Schoelkopf, Keiran Paster, Marco Dos Santos, Stephen Marcus McAleer, Albert Q. Jiang, Jia Deng, Stella Biderman, and Sean Welleck. Llemma: An open language model for mathematics. In *ICLR*, 2024.

Adrien Barbaresi. Trafilatura: A Web Scraping Library and Command-Line Tool for Text Discovery and Extraction. In *Proceedings of the Joint Conference of the 59th Annual Meeting of the Association for Computational Linguistics and the 11th International Joint Conference on Natural Language Processing: System Demonstrations*, pp. 122–131. Association for Computational Linguistics, 2021. URL `https://aclanthology.org/2021.acl-demo.15`.

Janek Bevendorff, Benno Stein, Matthias Hagen, and Martin Potthast. Elastic ChatNoir: Search Engine for the ClueWeb and the Common Crawl. In Leif Azzopardi, Allan Hanbury, Gabriella Pasi, and Benjamin Piwowarski (eds.), *Advances in Information Retrieval. 40th European Conference on IR Research (ECIR 2018)*, Lecture Notes in Computer Science, Berlin Heidelberg New York, March 2018. Springer.

Cody Blakeney, Mansheej Paul, Brett W Larsen, Sean Owen, and Jonathan Frankle. Does your data spark joy? performance gains from domain upsampling at the end of training. *arXiv preprint arXiv:2406.03476*, 2024.

Andrei Z Broder. Identifying and filtering near-duplicate documents. In *Annual symposium on combinatorial pattern matching*, pp. 1–10. Springer, 2000.

Mark Chen, Jerry Tworek, Heewoo Jun, Qiming Yuan, Henrique Ponde de Oliveira Pinto, Jared Kaplan, et al. Evaluating Large Language Models Trained on Code, 2021. URL `https://arxiv.org/abs/2107.03374`.

Karl Cobbe, Vineet Kosaraju, Mohammad Bavarian, Mark Chen, Heewoo Jun, Lukasz Kaiser, Matthias Plappert, Jerry Tworek, Jacob Hilton, Reiichiro Nakano, Christopher Hesse, and John Schulman. Training Verifiers to Solve Math Word Problems, 2021. URL `https://arxiv.org/abs/2110.14168`.

DeepSeek-AI, Aixin Liu, Bei Feng, Bing Xue, Bingxuan Wang, Bochao Wu, Chengda Lu, Chenggang Zhao, Chengqi Deng, Chenyu Zhang, Chong Ruan, Damai Dai, Daya Guo, Dejian Yang, Deli Chen, Dongjie Ji, Erhang Li, Fangyun Lin, Fucong Dai, Fuli Luo, Guangbo Hao, Guanting Chen, Guowei Li, H. Zhang, Han Bao, Hanwei Xu, Haocheng Wang, Haowei Zhang, Honghui Ding, Huajian Xin, Huazuo Gao, Hui Li, Hui Qu, J. L. Cai, Jian Liang, Jianzhong Guo, Jiaqi Ni, Jiashi Li, Jiawei Wang, Jin Chen, Jingchang Chen, Jingyang Yuan, Junjie Qiu, Junlong Li, Junxiao Song, Kai Dong, Kai Hu, Kaige Gao, Kang Guan, Kexin Huang, Kuai Yu, Lean Wang, Lecong Zhang, Lei Xu, Leyi Xia, Liang Zhao, Litong Wang, Liyue Zhang, Meng Li, Miaojun Wang, Mingchuan Zhang, Minghua Zhang, Minghui Tang, Mingming Li, Ning Tian, Panpan Huang, Peiyi Wang, Peng Zhang, Qiancheng Wang, Qihao Zhu, Qinyu Chen, Qiushi Du, R. J. Chen, R. L. Jin, Ruiqi Ge, Ruisong Zhang, Ruizhe Pan, Runji Wang, Runxin Xu, Ruoyu Zhang, Ruyi Chen, S. S. Li, Shanghao Lu, Shangyan Zhou, Shanhuang Chen, Shaoqing Wu, Shengfeng

Ye, Shengfeng Ye, Shirong Ma, Shiyu Wang, Shuang Zhou, Shuiping Yu, Shunfeng Zhou, Shuting Pan, T. Wang, Tao Yun, Tian Pei, Tianyu Sun, W. L. Xiao, Wangding Zeng, Wanjia Zhao, Wei An, Wen Liu, Wenfeng Liang, Wenjun Gao, Wenqin Yu, Wentao Zhang, X. Q. Li, Xiangyue Jin, Xianzu Wang, Xiao Bi, Xiaodong Liu, Xiaohan Wang, Xiaojin Shen, Xiaokang Chen, Xiaokang Zhang, Xiaosha Chen, Xiaotao Nie, Xiaowen Sun, Xiaoxiang Wang, Xin Cheng, Xin Liu, Xin Xie, Xingchao Liu, Xingkai Yu, Xinnan Song, Xinxia Shan, Xinyi Zhou, Xinyu Yang, Xinyuan Li, Xuecheng Su, Xuheng Lin, Y. K. Li, Y. Q. Wang, Y. X. Wei, Y. X. Zhu, Yang Zhang, Yanhong Xu, Yanhong Xu, Yanping Huang, Yao Li, Yao Zhao, Yaofeng Sun, Yaohui Li, Yaohui Wang, Yi Yu, Yi Zheng, Yichao Zhang, Yifan Shi, Yiliang Xiong, Ying He, Ying Tang, Yishi Piao, Yisong Wang, Yixuan Tan, Yiyang Ma, Yiyuan Liu, Yongqiang Guo, Yu Wu, Yuan Ou, Yuchen Zhu, Yuduan Wang, Yue Gong, Yuheng Zou, Yujia He, Yukun Zha, Yunfan Xiong, Yunxian Ma, Yuting Yan, Yuxiang Luo, Yuxiang You, Yuxuan Liu, Yuyang Zhou, Z. F. Wu, Z. Z. Ren, Zehui Ren, Zhangli Sha, Zhe Fu, Zhean Xu, Zhen Huang, Zhen Zhang, Zhenda Xie, Zhengyan Zhang, Zhewen Hao, Zhibin Gou, Zhicheng Ma, Zhigang Yan, Zhihong Shao, Zhipeng Xu, Zhiyu Wu, Zhongyu Zhang, Zhuoshu Li, Zihui Gu, Zijia Zhu, Zijun Liu, Zilin Li, Ziwei Xie, Ziyang Song, Ziyi Gao, and Zizheng Pan. Deepseek-v3 technical report, 2025. URL https://arxiv.org/abs/2412.19437.

Abhimanyu Dubey, Abhinav Jauhri, Abhinav Pandey, Abhishek Kadian, Ahmad Al-Dahle, Aiesha Letman, Akhil Mathur, Alan Schelten, Amy Yang, Angela Fan, et al. The llama 3 herd of models. *arXiv preprint arXiv:2407.21783*, 2024.

István Endrédy and Attila Novák. More effective boilerplate removal-the goldminer algorithm. *Polibits*, 48:79–83, 12 2013. doi: 10.17562/PB-48-10.

Leo Gao, Stella Biderman, Sid Black, Laurence Golding, Travis Hoppe, Charles Foster, Jason Phang, Horace He, Anish Thite, Noa Nabeshima, et al. The pile: An 800gb dataset of diverse text for language modeling. *arXiv preprint arXiv:2101.00027*, 2020.

Daya Guo, Dejian Yang, Haowei Zhang, Junxiao Song, Ruoyu Zhang, Runxin Xu, Qihao Zhu, Shirong Ma, Peiyi Wang, Xiao Bi, et al. Deepseek-r1: Incentivizing reasoning capability in llms via reinforcement learning. *arXiv preprint arXiv:2501.12948*, 2025.

Xiaotian Han, Yiren Jian, Xuefeng Hu, Haogeng Liu, Yiqi Wang, Qihang Fan, Yuang Ai, Huaibo Huang, Ran He, Zhenheng Yang, and Quanzeng You. Infimm-webmath-40b: Advancing multi-modal pre-training for enhanced mathematical reasoning, 2024. URL https://arxiv.org/abs/2409.12568.

Dan Hendrycks, Collin Burns, Steven Basart, Andy Zou, Mantas Mazeika, Dawn Song, and Jacob Steinhardt. Measuring massive multitask language understanding, 2021a. URL https://arxiv.org/abs/2009.03300.

Dan Hendrycks, Collin Burns, Saurav Kadavath, Akul Arora, Steven Basart, Eric Tang, Dawn Song, and Jacob Steinhardt. Measuring mathematical problem solving with the math dataset. *NeurIPS*, 2021b.

Shengding Hu, Yuge Tu, Xu Han, Chaoqun He, Ganqu Cui, Xiang Long, Zhi Zheng, Yewei Fang, Yuxiang Huang, Weilin Zhao, et al. Minicpm: Unveiling the potential of small language models with scalable training strategies. *arXiv preprint arXiv:2404.06395*, 2024.

Binyuan Hui, Jian Yang, Zeyu Cui, Jiaxi Yang, Dayiheng Liu, Lei Zhang, Tianyu Liu, Jiajun Zhang, Bowen Yu, Keming Lu, Kai Dang, Yang Fan, Yichang Zhang, An Yang, Rui Men, Fei Huang, Bo Zheng, Yibo Miao, Shanghaoran Quan, Yunlong Feng, Xingzhang Ren, Xuancheng Ren, Jingren Zhou, and Junyang Lin. Qwen2.5-coder technical report, 2024. URL https://arxiv.org/abs/2409.12186.

Denis Kocetkov, Raymond Li, Loubna Ben Allal, Jia Li, Chenghao Mou, Carlos Muñoz Ferrandis, Yacine Jernite, Margaret Mitchell, Sean Hughes, Thomas Wolf, et al. The stack: 3 tb of permissively licensed source code. *arXiv preprint arXiv:2211.15533*, 2022.

Vijay Korthikanti, Jared Casper, Sangkug Lym, Lawrence McAfee, Michael Andersch, Mohammad Shoeybi, and Bryan Catanzaro. Reducing Activation Recomputation in Large Transformer Models, 2022. URL https://arxiv.org/abs/2205.05198.

Katherine Lee, Daphne Ippolito, Andrew Nystrom, Chiyuan Zhang, Douglas Eck, Chris Callison-Burch, and Nicholas Carlini. Deduplicating training data makes language models better. In *ACL*, pp. 8424–8445, 2022.

Aitor Lewkowycz, Anders Andreassen, David Dohan, Ethan Dyer, Henryk Michalewski, Vinay Ramasesh, Ambrose Slone, Cem Anil, Imanol Schlag, Theo Gutman-Solo, et al. Solving quantitative reasoning problems with language models. *NeurIPS*, 35:3843–3857, 2022.

Raymond Li, Loubna Ben Allal, Yangtian Zi, Niklas Muennighoff, Denis Kocetkov, Chenghao Mou, Marc Marone, Christopher Akiki, Jia Li, Jenny Chim, et al. Starcoder: may the source be with you! *arXiv preprint arXiv:2305.06161*, 2023.

Hunter Lightman, Vineet Kosaraju, Yura Burda, Harrison Edwards, Bowen Baker, Teddy Lee, Jan Leike, John Schulman, Ilya Sutskever, and Karl Cobbe. Let's verify step by step. *CoRR*, abs/2305.20050, 2023. doi: 10.48550/arXiv.2305.20050. URL https://doi.org/10.48550/arXiv.2305.20050.

Aixin Liu, Bei Feng, Bing Xue, Bingxuan Wang, Bochao Wu, Chengda Lu, Chenggang Zhao, Chengqi Deng, Chenyu Zhang, Chong Ruan, et al. Deepseek-v3 technical report. *arXiv preprint arXiv:2412.19437*, 2024.

Jiawei Liu, Chunqiu Steven Xia, Yuyao Wang, and Lingming Zhang. Is Your Code Generated by ChatGPT Really Correct? Rigorous Evaluation of Large Language Models for Code Generation. *arXiv preprint arXiv:2305.01210*, 2023. doi: https://doi.org/10.48550/arXiv.2305.01210. URL https://arxiv.org/abs/2305.01210.

Ilya Loshchilov and Frank Hutter. Decoupled weight decay regularization. *arXiv preprint arXiv:1711.05101*, 2017.

Anton Lozhkov, Raymond Li, Loubna Ben Allal, Federico Cassano, Joel Lamy-Poirier, Nouamane Tazi, Ao Tang, Dmytro Pykhtar, Jiawei Liu, Yuxiang Wei, et al. Starcoder 2 and the stack v2: The next generation. *arXiv preprint arXiv:2402.19173*, 2024.

Niklas Muennighoff, Alexander M. Rush, Boaz Barak, Teven Le Scao, Aleksandra Piktus, Nouamane Tazi, Sampo Pyysalo, Thomas Wolf, and Colin Raffel. Scaling data-constrained language models, 2023.

NVIDIA, :, Aaron Blakeman, Aarti Basant, Abhinav Khattar, Adithya Renduchintala, Akhiad Bercovich, Aleksander Ficek, Alexis Bjorlin, Ali Taghibakhshi, Amala Sanjay Deshmukh, Ameya Sunil Mahabaleshwarkar, Andrew Tao, Anna Shors, Ashwath Aithal, Ashwin Poojary, Ayush Dattagupta, Balaram Buddharaju, Bobby Chen, Boris Ginsburg, Boxin Wang, Brandon Norick, Brian Butterfield, Bryan Catanzaro, Carlo del Mundo, Chengyu Dong, Christine Harvey, Christopher Parisien, Dan Su, Daniel Korzekwa, Danny Yin, Daria Gitman, David Mosallanezhad, Deepak Narayanan, Denys Fridman, Dima Rekesh, Ding Ma, Dmytro Pykhtar, Dong Ahn, Duncan Riach, Dusan Stosic, Eileen Long, Elad Segal, Ellie Evans, Eric Chung, Erick Galinkin, Evelina Bakhturina, Ewa Dobrowolska, Fei Jia, Fuxiao Liu, Gargi Prasad, Gerald Shen, Guilin Liu, Guo Chen, Haifeng Qian, Helen Ngo, Hongbin Liu, Hui Li, Igor Gitman, Ilia Karmanov, Ivan Moshkov, Izik Golan, Jan Kautz, Jane Polak Scowcroft, Jared Casper, Jarno Seppanen, Jason Lu, Jason Sewall, Jiaqi Zeng, Jiaxuan You, Jimmy Zhang, Jing Zhang, Jining Huang, Jinze Xue, Jocelyn Huang, Joey Conway, John Kamalu, Jon Barker, Jonathan Cohen, Joseph Jennings, Jupinder Parmar, Karan Sapra, Kari Briski, Kateryna Chumachenko, Katherine Luna, Keshav Santhanam, Kezhi Kong, Kirthi Sivamani, Krzysztof Pawelec, Kumar Anik, Kunlun Li, Lawrence McAfee, Leon Derczynski, Lindsey Pavao, Luis Vega, Lukas Voegtle, Maciej Bala, Maer Rodrigues de Melo, Makesh Narsimhan Sreedhar, Marcin Chochowski, Markus Kliegl, Marta Stepniewska-Dziubinska, Matthieu Le, Matvei Novikov, Mehrzad Samadi, Michael Andersch, Michael Evans, Miguel Martinez, Mike Chrzanowski, Mike Ranzinger, Mikolaj Blaz, Misha Smelyanskiy, Mohamed Fawzy, Mohammad Shoeybi, Mostofa Patwary, Nayeon Lee, Nima Tajbakhsh, Ning Xu, Oleg Rybakov, Oleksii Kuchaiev, Olivier Delalleau, Osvald Nitski, Parth Chadha, Pasha Shamis, Paulius Micikevicius, Pavlo Molchanov, Peter Dykas, Philipp Fischer, Pierre-Yves Aquilanti, Piotr Bialecki, Prasoon Varshney, Pritam Gundecha, Przemek Tredak, Rabeeh Karimi, Rahul Kandu, Ran El-Yaniv, Raviraj Joshi, Roger Waleffe, Ruoxi Zhang, Sabrina

Kavanaugh, Sahil Jain, Samuel Kriman, Sangkug Lym, Sanjeev Satheesh, Saurav Muralidharan, Sean Narenthiran, Selvaraj Anandaraj, Seonmyeong Bak, Sergey Kashirsky, Seungju Han, Shantanu Acharya, Shaona Ghosh, Sharath Turuvekere Sreenivas, Sharon Clay, Shelby Thomas, Shrimai Prabhumoye, Shubham Pachori, Shubham Toshniwal, Shyamala Prayaga, Siddhartha Jain, Sirshak Das, Slawek Kierat, Somshubra Majumdar, Song Han, Soumye Singhal, Sriharsha Niverty, Stefania Alborghetti, Suseella Panguluri, Swetha Bhendigeri, Syeda Nahida Akter, Szymon Migacz, Tal Shiri, Terry Kong, Timo Roman, Tomer Ronen, Trisha Saar, Tugrul Konuk, Tuomas Rintamaki, Tyler Poon, Ushnish De, Vahid Noroozi, Varun Singh, Vijay Korthikanti, Vitaly Kurin, Wasi Uddin Ahmad, Wei Du, Wei Ping, Wenliang Dai, Wonmin Byeon, Xiaowei Ren, Yao Xu, Yejin Choi, Yian Zhang, Ying Lin, Yoshi Suhara, Zhiding Yu, Zhiqi Li, Zhiyu Li, Zhongbo Zhu, Zhuolin Yang, and Zijia Chen. Nemotron-h: A family of accurate and efficient hybrid mamba-transformer models, 2025. URL https://arxiv.org/abs/2504.03624.

Team OLMo, Pete Walsh, Luca Soldaini, Dirk Groeneveld, Kyle Lo, Shane Arora, Akshita Bhagia, Yuling Gu, Shengyi Huang, Matt Jordan, et al. 2 olmo 2 furious. *arXiv preprint arXiv:2501.00656*, 2024.

OpenAI. Learning to reason with llms, 2024. URL https://openai.com/index/learning-to-reason-with-llms/.

Keiran Paster, Marco Dos Santos, Zhangir Azerbayev, and Jimmy Ba. Openwebmath: An open dataset of high-quality mathematical web text. *ICLR*, 2024.

Guilherme Penedo, Quentin Malartic, Daniel Hesslow, Ruxandra Cojocaru, Hamza Alobeidli, Alessandro Cappelli, Baptiste Pannier, Ebtesam Almazrouei, and Julien Launay. The refined-web dataset for falcon llm: Outperforming curated corpora with web data only. *NeurIPS*, 36: 79155–79172, 2023.

Stanislas Polu and Ilya Sutskever. Generative language modeling for automated theorem proving. *arXiv preprint arXiv:2009.03393*, 2020.

Qwen, :, An Yang, Baosong Yang, Beichen Zhang, Binyuan Hui, Bo Zheng, Bowen Yu, Chengyuan Li, Dayiheng Liu, Fei Huang, Haoran Wei, Huan Lin, Jian Yang, Jianhong Tu, Jianwei Zhang, Jianxin Yang, Jiaxi Yang, Jingren Zhou, Junyang Lin, Kai Dang, Keming Lu, Keqin Bao, Kexin Yang, Le Yu, Mei Li, Mingfeng Xue, Pei Zhang, Qin Zhu, Rui Men, Runji Lin, Tianhao Li, Tianyi Tang, Tingyu Xia, Xingzhang Ren, Xuancheng Ren, Yang Fan, Yang Su, Yichang Zhang, Yu Wan, Yuqiong Liu, Zeyu Cui, Zhenru Zhang, and Zihan Qiu. Qwen2.5 technical report, 2025. URL https://arxiv.org/abs/2412.15115.

Samyam Rajbhandari, Jeff Rasley, Olatunji Ruwase, and Yuxiong He. ZeRO: Memory Optimizations Toward Training Trillion Parameter Models, 2020. URL https://arxiv.org/abs/1910.02054.

Zhihong Shao, Peiyi Wang, Qihao Zhu, Runxin Xu, Junxiao Song, Xiao Bi, Haowei Zhang, Mingchuan Zhang, YK Li, Y Wu, et al. Deepseekmath: Pushing the limits of mathematical reasoning in open language models. *arXiv preprint arXiv:2402.03300*, 2024.

Mohammad Shoeybi, Mostofa Patwary, Raul Puri, Patrick LeGresley, Jared Casper, and Bryan Catanzaro. Megatron-LM: Training Multi-Billion Parameter Language Models Using Model Parallelism, 2020. URL https://arxiv.org/abs/1909.08053.

Dan Su, Kezhi Kong, Ying Lin, Joseph Jennings, Brandon Norick, Markus Kliegl, Mostofa Patwary, Mohammad Shoeybi, and Bryan Catanzaro. Nemotron-CC: Transforming Common Crawl into a refined long-horizon pretraining dataset. In Wanxiang Che, Joyce Nabende, Ekaterina Shutova, and Mohammad Taher Pilehvar (eds.), *Proceedings of the 63rd Annual Meeting of the Association for Computational Linguistics (Volume 1: Long Papers)*, pp. 2459–2475, Vienna, Austria, July 2025. Association for Computational Linguistics. ISBN 979-8-89176-251-0. URL https://aclanthology.org/2025.acl-long.123/.

Liping Tang, Nikhil Ranjan, Omkar Pangarkar, Xuezhi Liang, Zhen Wang, Li An, Bhaskar Rao, Linghao Jin, Huijuan Wang, Zhoujun Cheng, Suqi Sun, Cun Mu, Victor Miller, Xuezhe Ma, Yue Peng, Zhengzhong Liu, and Eric P. Xing. Txt360: A top-quality llm pre-training dataset requires the perfect blend, 2024. URL https://huggingface.co/spaces/LLM360/TxT360.

Qwen Team. Qwen2 technical report. *arXiv preprint arXiv:2407.10671*, 2024.

Yury Tokpanov, Paolo Glorioso, Quentin Anthony, and Beren Millidge. Zyda-2: a 5 trillion token high-quality dataset. *arXiv preprint arXiv:2411.06068*, 2024.

Yubo Wang, Xueguang Ma, Ge Zhang, Yuansheng Ni, Abhranil Chandra, Shiguang Guo, Weiming Ren, Aaran Arulraj, Xuan He, Ziyan Jiang, Tianle Li, Max Ku, Kai Wang, Alex Zhuang, Rongqi Fan, Xiang Yue, and Wenhu Chen. Mmlu-pro: A more robust and challenging multi-task language understanding benchmark, 2024a. URL https://arxiv.org/abs/2406.01574.

Zengzhi Wang, Xuefeng Li, Rui Xia, and Pengfei Liu. Mathpile: A billion-token-scale pretraining corpus for math. *NeurIPS*, 37:25426–25468, 2024b.

Sean Welleck, Jiacheng Liu, Ronan Le Bras, Hannaneh Hajishirzi, Yejin Choi, and Kyunghyun Cho. Naturalproofs: Mathematical theorem proving in natural language. *arXiv preprint arXiv:2104.01112*, 2021.

An Yang, Beichen Zhang, Binyuan Hui, Bofei Gao, Bowen Yu, Chengpeng Li, Dayiheng Liu, Jianhong Tu, Jingren Zhou, Junyang Lin, et al. Qwen2.5-math technical report: Toward mathematical expert model via self-improvement. *arXiv preprint arXiv:2409.12122*, 2024.

Shuo Yang, Wei-Lin Chiang, Lianmin Zheng, Joseph E. Gonzalez, and Ion Stoica. Rethinking benchmark and contamination for language models with rephrased samples, 2023.

Fan Zhou, Zengzhi Wang, Nikhil Ranjan, Zhoujun Cheng, Liping Tang, Guowei He, Zhengzhong Liu, and Eric P Xing. Megamath: Pushing the limits of open math corpora. *arXiv preprint arXiv:2504.02807*, 2025.

# A    APPENDIX

## A.1    USE OF LARGE LANGUAGE MODELS (LLMS)

We used a large language model (LLM) solely to aid in polishing the writing of this paper, including improving grammar, clarity, and flow. The LLM was not involved in the research ideation, methodology, experiments, analysis, or interpretation of results.

## A.2    PROMPT USED FOR TOPIC CLASSIFICATION

We employ the following prompt to classify documents into a predefined set of categories. During classification, the large language model (LLM) occasionally produces category labels that fall outside the predefined taxonomy: mathematics, computer science, physics, statistics, economics, chemistry, and other. To maintain consistency and reduce label fragmentation, any out-of-taxonomy label is reassigned to the category other, ensuring a coherent and structured category distribution.

> You are a topic classification assistant.
> Given the following document text, identify its main topic from this list only:
> – Mathematics
> – Computer Science
> – Physics
> – Statistics
> – Chemistry
> – Economics
> – Other
>
> Choose the single most relevant category from the list.
> Document:
> {text}
>
> Your output should be only 1 word. Finish your response right after category and do not add any explanation.

## A.3    QUALITATIVE EXAMPLES

This section presents qualitative comparisons among OpenWebMath (OWM), MegaMath-Pro, FineMath-4+, and Nemotron-CC-Math-4+, highlighting differences in content quality.

### A.3.1    DEGENERATE CASES IN MEGAMATH-PRO DATASET

We identified a subset of degenerate generations within the MegaMath-Pro dataset. Representative examples are presented below to illustrate this phenomenon. Notably, these samples achieve unexpectedly high scores on both mathematical and language scores, raising concerns about the dataset's overall reliability for pretraining LLMs. For each example, we provide the associated metadata. The excerpts shown correspond to the initial portion of each generation; in every case, the text extends over several additional pages, repeating the final sentence displayed.

### A.3.2    SIDE BY SIDE COMPARISON BETWEEN OUR DATASET AND PRIOR WORK

We observe that our pipeline not only keep the math equations but also keep the codes and their formatting. We observe that previous pipeline in most cases are not keeping codes or lose their formatting. This is specifically important for languages like python. To highlight this difference, we provide two sets of examples demonstrating both code and mathematical equations. Notably, inline equations are often removed in prior work, such as MegaMath.

---

A degenerate sample from MegaMath-Pro

---

The Integral Calculator is able to calculate integrals online of the composition of common functions, using integral properties, the different mechanisms of integration and calculation online. The Integral Calculator is a simple online calculator that computes the definite and indefinite integrals. The Integral Calculator will show you a graphical version of your input while you type.
The Integral Calculator is a free online tool for calculating the value of a definite integral. The Integral Calculator, part of the graphing calculator, helps with one variable calculus. The Integral Calculator supports definite and indefinite integrals (antiderivatives) as well as integrating functions with many variables.
The Integral Calculator is able to calculate integrals online of the composition of common functions, using integral properties, the different mechanisms of integration and calculation online. The Integral Calculator is a simple online calculator that computes the definite and indefinite integrals. The Integral Calculator will show you a graphical version of your input while you type.
The Integral Calculator is a free online tool for calculating the value of a definite integral. The Integral Calculator, part of the graphing calculator, helps with one variable calculus. The Integral Calculator supports definite and indefinite integrals (antiderivatives) as well as integrating functions with many variables.
The Integral Calculator is able to calculate integrals online of the composition of common functions, using integral properties, the different mechanisms of integration and calculation online. The Integral Calculator is a simple online calculator that computes the definite and indefinite integrals. The Integral Calculator will show you a graphical version of your input while you type.
The Integral Calculator is a free online tool for calculating the value of a definite integral. The Integral Calculator, part of the graphing calculator, helps with one variable calculus. The Integral Calculator supports definite and indefinite integrals (antiderivatives) as well as integrating functions with many variables.
The Integral Calculator is able to calculate integrals online of the composition of common functions, using integral properties, the different mechanisms of integration and calculation online. The Integral Calculator is a simple online calculator that computes the definite and indefinite integrals. The Integral Calculator will show you a graphical version of your input while you type.
The Integral Calculator is a free online tool for calculating the value of a definite integral. The Integral Calculator, part of the graphing calculator, helps with one variable calculus. The Integral Calculator supports definite and indefinite integrals (antiderivatives) as well as integrating functions with many variables.
The Integral Calculator is able to calculate integrals online of the composition of common functions, using integral properties, the different mechanisms of integration and calculation online. The Integral Calculator is a simple online calculator that computes the definite and indefinite integrals. The Integral Calculator will show you a graphical version of your input while you type.
The Integral Calculator is a free online tool for calculating the value of a definite integral. The Integral Calculator, part of the graphing calculator, helps with one variable calculus. The Integral Calculator supports definite and indefinite integrals (antiderivatives) as well as integrating functions with many variables.
The Integral Calculator is able to calculate integrals online of the composition of common functions, using integral properties, the different mechanisms of integration and calculation online. The Integral Calculator is a simple online calculator that computes the definite and indefinite integrals. The Integral Calculator will show you a graphical version of your input while you type.
The Integral Calculator is a free online tool for calculating the value of a definite integral.

---

**Meta:**

**URL:** http://031c82c.netsolhost.com/yvcmr4/article.php?c08ee4=complex-integration-calculator

**Math Score:** 0.9996713399887085

**Lang Score:** 0.7893766164779663

**WARC Filename:** CC-MAIN-2022-21/segments/1652662531762.30/warc/CC-MAIN-20220520061824-00605.warc.gz

A degenerate sample from MegaMath-Pro

The angle will be calculated and displayed. Use the law of cosines to find one of the angles. The angle will be calculated and displayed. The angle will be calculated and displayed.
The angle will be calculated and displayed. Use the law of cosines to find one of the angles. The angle will be calculated and displayed. The angle will be calculated and displayed.
The angle will be calculated and displayed. Use the law of cosines to find one of the angles. The angle will be calculated and displayed. The angle will be calculated and displayed.
The angle will be calculated and displayed. Use the law of cosines to find one of the angles. The angle will be calculated and displayed. The angle will be calculated and displayed.
The angle will be calculated and displayed. Use the law of cosines to find one of the angles. The angle will be calculated and displayed. The angle will be calculated and displayed.
The angle will be calculated and displayed. Use the law of cosines to find one of the angles. The angle will be calculated and displayed. The angle will be calculated and displayed.
The angle will be calculated and displayed. Use the law of cosines to find one of the angles. The angle will be calculated and displayed. The angle will be calculated and displayed.
The angle will be calculated and displayed. Use the law of cosines to find one of the angles. The angle will be calculated and displayed. The angle will be calculated and displayed.
The angle will be calculated and displayed. Use the law of cosines to find one of the angles. The angle will be calculated and displayed. The angle will be calculated and displayed.
The angle will be calculated and displayed. Use the law of cosines to find one of the angles. The angle will be calculated and displayed. The angle will be calculated and displayed.
The angle will be calculated and displayed. Use the law of cosines to find one of the angles. The angle will be calculated and displayed. The angle will be calculated and displayed.
The angle will be calculated and displayed. Use the law of cosines to find one of the angles. The angle will be calculated and displayed. The angle will be calculated and displayed.
The angle will be calculated and displayed. Use the law of cosines to find one of the angles. The angle will be calculated and displayed. The angle will be calculated and displayed.
The angle will be calculated and displayed. Use the law of cosines to find one of the angles. The angle will be calculated and displayed. The angle will be calculated and displayed.
The angle will be calculated and displayed. Use the law of cosines to find one of the angles. The angle will be calculated and displayed. The angle will be calculated and displayed.
The angle will be calculated and displayed. Use the law of cosines to find one of the angles. The angle will be calculated and displayed. The angle will be calculated and displayed.
The angle will be calculated and displayed. Use the law of cosines to find one of the angles. The angle will be calculated and displayed. The angle will be calculated and displayed.
The angle will be calculated and displayed. Use the law of cosines to find one of the angles. The angle will be calculated and displayed. The angle will be calculated and displayed.
The angle will be calculated and displayed. Use the law of cosines to find one of the angles. The angle will be calculated and displayed. The angle will be calculated and displayed.
The angle will be calculated and displayed. Use the law of cosines to find one of the angles. The angle will be calculated and displayed. The angle will be calculated and displayed.
The angle will be calculated and displayed. Use the law of cosines to find one of the angles. The angle will be calculated and displayed. The angle will be calculated and displayed.
The angle will be calculated and displayed. Use the law of cosines to find one of the angles. The angle will be calculated and displayed. The angle will be calculated and displayed.
The angle will be calculated and displayed. Use the law of cosines to find one of the angles. The angle will be calculated and displayed. The angle will be calculated and displayed.

**Meta:**

---

A degenerate sample from MegaMath-Pro

---

The equation of the axis of symmetry in a vertical parabola is equal to the x-coordinate of the vertex. The axis of symmetry always passes through the vertex of the parabola. The x-coordinate of the vertex is equal to the formula.

To learn about the axis of symmetry, watch this tutorial! The axis of symmetry is the line that divides the graph into two perfect halves. The axis of symmetry is always a vertical line of the form x = n, where n is a real number. A parabola is the graph of a quadratic function. Each parabola has a line of symmetry. Also known as the axis of symmetry, this line divides the parabola into mirror images. The line of symmetry is always a vertical line of the form x = n, where n is a real number.

When graphing, we want to include certain special points in the graph. The y-intercept is the point where the graph intersects the y-axis. The x-intercepts are the points where the graph intersects the x-axis. The vertex is the point that defines the minimum or maximum of the graph.

The axis of symmetry for an equation with x 2 is the vertical line that passes through the vertex. The axis of symmetry is the line x = h, where (h, k) is the vertex of the parabola.

The axis of symmetry is the line that divides the graph into two perfect halves. The axis of symmetry is always a vertical line of the form x = n, where n is a real number.

The equation of the axis of symmetry in a vertical parabola is equal to the x-coordinate of the vertex. The axis of symmetry always passes through the vertex of the parabola. The x-coordinate of the vertex is equal to the formula.

To learn about the axis of symmetry, watch this tutorial! The axis of symmetry is the line that divides the graph into two perfect halves. The axis of symmetry is always a vertical line of the form x = n, where n is a real number.

A parabola is the graph of a quadratic function. Each parabola has a line of symmetry. Also known as the axis of symmetry, this line divides the parabola into mirror images. The line of symmetry is always a vertical line of the form x = n, where n is a real number.

When graphing, we want to include certain special points in the graph. The y-intercept is the point where the graph intersects the y-axis. The x-intercepts are the points where the graph intersects the x-axis. The vertex is the point that defines the minimum or maximum of the graph.

The axis of symmetry for an equation with x 2 is the vertical line that passes through the vertex. The axis of symmetry is the line x = h, where (h, k) is the vertex of the parabola.

The axis of symmetry is the line that divides the graph into two perfect halves. The axis of symmetry is always a vertical line of the form x = n, where n is a real number.

The equation of the axis of symmetry in a vertical parabola is equal to the x-coordinate of the vertex. The axis of symmetry always passes through the vertex of the parabola. The x-coordinate of the vertex is equal to the formula.

To learn about the axis of symmetry, watch this tutorial! The axis of symmetry is the line that divides the graph into two perfect halves. The axis of symmetry is always a vertical line of the form x = n, where n is a real number.

A parabola is the graph of a quadratic function. Each parabola has a line of symmetry. Also known as the axis of symmetry, this line divides the parabola into mirror images. The line of symmetry is always a vertical line of the form x = n, where n is a real number.

When graphing, we want to include certain special points in the graph. The y-intercept is the point where the graph intersects the y-axis. The x-intercepts are the points where the graph intersects the x-axis. The vertex is the point that defines the minimum or maximum of the graph.

The axis of symmetry for an equation with x 2 is the vertical line that passes through the vertex. The axis of symmetry is the line x = h, where (h, k) is the vertex of the parabola.

The axis of symmetry is the line that divides the graph into two perfect halves. The axis of symmetry is always a vertical line of the form x = n, where n is a real number. The equation of the axis of symmetry in a vertical parabola is equal to the x-coordinate of the vertex.

---

**Meta:**

**URL:** http://1798091312.srv040122.webreus.net/q8oh94/c5d31c-vertical-symmetry-graph

**Math Score:** 0.9988757371902466

**Lang Score:** 0.9151933789253235

**WARC Filename:** CC-MAIN-2021-25/segments/1623488273983.63/warc/CC-MAIN-20210621120456-00277.warc.gz

---

**Lynx output**

#10000 Terabyte

  10000 Terabyte

  about opensource disclaimer

  (BUTTON)
  about opensource disclaimer

Detailed explanation of a smart solution to an algo problem beating 99.9% submission

  Written on January 7th, 2018 by @10000TB
  [attachments_article_algorithm_col_slide_lamparas−colgantes−algorithm−slide−03.jpg]

  This post is about a coding problem and why the solution I pasted down below is smart.
  Problem:
Given two sparse matrices A and B, return the result of AB.

You may assume that A's column number is equal to B's row number.

Example:

A = [
 [ 1, 0, 0],
 [−1, 0, 3]
]

B = [
 [ 7, 0, 0 ],
 [ 0, 0, 0 ],
 [ 0, 0, 1 ]
]

   | 1 0 0 |  | 7 0 0 |  | 7 0 0 |
AB = | −1 0 3 | x | 0 0 0 | = | −7 0 3 |
           | 0 0 1 |

  If it is of your interest, I would recommend you take a few minutes to think about how you would
     approach this problem!

  The main focus of this post is to 1)explain in detail why the provided solution is smart and 2)make
     some improvements/tweaks in the code of the smart solution to show you which part is really
     essential, 3) also i will briefly mention why Sparse Matrix Manipulation can help make some
     improvements on top of the smart solution.
  a) Originally, the normal way to calculate the multiplication of two metrics A, and B is as follow: We
     take the the all values from the first line of A, and all values from the first column of B, and
     multiply the corresponding values and sum them up, the final sum is the value for the location of
     first column and first row in final result matrix. Similarly, the value at [ i ][ j ] of result matrix C,
     which is C[ i ][ j ] is calculated as:
  C[ i ][ j ] = A[ i ][0]B[0][j] + A[i][1]B[1][j] + A[i][2]B[2][j] + ... A[i][K]B[K][j]
  (which is the sum of each multiplication of corresponding K values from row i of A and K values from
     column j of B)
  The Key is: if we calculate it this way, we finishing calculating the final value for the result matrix at
     once
  Then a brute force solution is as follow:

```
public class Solution {
    public int[][] multiply(int[][] A, int[][] B) {
        int m = A.length, n = A[0].length, nB = B[0].length;
        int[][] C = new int[m][nB];

        for(int i = 0; i < m; i++) {
            for (int j = 0; j < nB; j++) {
                for(int k = 0; k < n; k++) {
                    C[i][j] += A[i][k] * B[k][j];
                }
            }
        }
        return C;
    }
}
```

b) The smart solution: the key part of smart solution is that: it does not calculate the final result at once, and it takes each value from A, and calculate and partial sum and accumulate it into the final spot:

For example, for each value A[i][k], if it is not zero, it will be used at most nB times ( n is B[0].length ), which can be illustrated as follow: Generally for the following equation:

C[i][j] = A[i][0]B[0][j] + A[i][1]B[1][j] + A[i][2]B[2][j] + ... A[i][k]B[k][j] .... A[i][K]B[K][j]

j can be from 0 to nB, if we write all of them down, it will like following:

For i from 0 to nB:

C[ i ][ 0 ]=A[ i ][0]B[0][0] + A[i][1]B[1][0] + A[i][2]B[2][0] + ... A[i][k]B[k][0] .... A[i][K]B[K][0]

C[ i ][ 1 ]=A[ i ][0]B[0][1] + A[i][1]B[1][1] + A[i][2]B[2][1] + ... A[i][k]B[k][0] .... A[i][K]B[K][1]

...

C[ i ][ nB ]=A[ i ][0]B[0][nB] + A[i][1]B[1][nB] + A[i][2]B[2][nB] + ... A[i][k]B[k][nB] .... A[i][K]*B[K][nB]

As you can see from above: for the same value A[i][k] from the first matrix, it will be used at most nB times if A[i][k] is not zero. And the smart solution is taking advantage of that!!!, the smart solution can be described as:

For each value A[i][k] in matrix A, if it is not zero, we calculate A[i][k] * B[k][j] and accumulate it into C[ i ][ j ] (Key part: the C[ i ][ j ] by now is not the final value in the result matrix !! Remember, in the brute force solution, the final value of C[i][j], takes sum of all multiplication values of K corresponding values from A and B? here C[ i ][ j ] is only sum of some multiplication values, NOT ALL until the program is done)

BY NOW, it is very clear that, if the value A[ i ][ k ] from matrix is zero, we skip a For−loop− calculation, which is a loop iterating nB times, and this is the key part of why the smart solution is smart!!!

The smart solution code is as follow:

```
public class Solution {
    public int[][] multiply(int[][] A, int[][] B) {
        int m = A.length, n = A[0].length, nB = B[0].length;
        int[][] C = new int[m][nB];

        for(int i = 0; i < m; i++) {
            for(int k = 0; k < n; k++) {
                if (A[i][k] != 0) {
                    for (int j = 0; j < nB; j++) {
                        if (B[k][j] != 0) C[i][j] += A[i][k] * B[k][j];
                    }
                }
            }
        }
        return C;
    }
}
```

(Credit:@yavinci; I am having a different version of the solution, so I am directly referencing the original version as a reference to demonstrate how mine is different).

Based on the discussion above, the inner checking (if (B[k][j] != 0)) is actually not necessary, because whether or not we have that check, we still iterate nB times, ( since the operation C[i][j] += A[i][k] * B[k][j]; inside the if−check is O(1) time)

So the smart solution can also be written as follow by removing the check ( which is my version ):

```
public class Solution {
  public int[][] multiply(int[][] A, int[][] B) {
    int m = A.length, n = A[0].length, nB = B[0].length;
    int[][] C = new int[m][nB];

    for(int i = 0; i < m; i++) {
      for(int k = 0; k < n; k++) {
        if (A[i][k] != 0) {
          for (int j = 0; j < nB; j++) {
            if (B[k][j] != 0) C[i][j] += A[i][k] * B[k][j];
          }
        }
      }
    }
    return C;
  }
}
```

c) "Sparse matrix manipultion" helps, if we compress the first sparse matrix into rows of lists( in each row list, it contains ( value, index ) pair ), we actually don't need to go over all values in a row in matrix A when are calculating the final result matrix. But Overall, it does not help improve run time algorithmatically!!

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

A sample from Nemotron-CC-Math

# 10000 Terabyte

## Detailed Explanation of a Smart Solution to an Algorithm Problem

*Written on January 7th, 2018 by @10000TB*

This post is about a coding problem and why the solution provided is smart. The problem is as follows:

**Problem:** Given two sparse matrices \( A \) and \( B \), return the result of \( AB \). You may assume that \( A \)'s column number is equal to \( B \)'s row number.

**Example:**

\[
A = \begin{bmatrix}
1 & 0 & 0 \\
−1 & 0 & 3
\end{bmatrix}
\]

\[
B = \begin{bmatrix}
7 & 0 & 0 \\
0 & 0 & 0 \\
0 & 0 & 1
\end{bmatrix}
\]

\[
AB = \begin{bmatrix}
1 & 0 & 0 \\
−1 & 0 & 3
\end{bmatrix}
\times
\begin{bmatrix}
7 & 0 & 0 \\
0 & 0 & 0 \\
0 & 0 & 1
\end{bmatrix}
=
\begin{bmatrix}
7 & 0 & 0 \\
−7 & 0 & 3
\end{bmatrix}
\]

If you are interested, consider how you would approach this problem!

The main focus of this post is to:
1. Explain in detail why the provided solution is smart.
2. Make some improvements/tweaks in the code of the smart solution to show which part is really essential.
3. Briefly mention why Sparse Matrix Manipulation can help make improvements.

### a) Brute Force Solution

The normal way to calculate the multiplication of two matrices \( A \) and \( B \) is as follows: We take all values from the first row of \( A \) and all values from the first column of \( B \), multiply the corresponding values and sum them up. The final sum is the value for the location of the first row and first column in the final result matrix \( C \). Similarly, the value at \( C[i][j] \) is calculated as:

\[
C[i][j] = A[i][0]B[0][j] + A[i][1]B[1][j] + A[i][2]B[2][j] + \ldots + A[i][K]B[K][j]
\]

The brute force solution is as follows:

```java
public class Solution {
    public int[][] multiply(int[][] A, int[][] B) {
        int m = A.length, n = A[0].length, nB = B[0].length;
        int[][] C = new int[m][nB];

        for (int i = 0; i < m; i++) {
            for (int j = 0; j < nB; j++) {
                for (int k = 0; k < n; k++) {
                    C[i][j] += A[i][k] * B[k][j];
                }
            }
        }
        return C;
    }
}
```

### b) The Smart Solution

The key part of the smart solution is that it does not calculate the final result at once. Instead, it takes each value from \( A \), calculates a partial sum, and accumulates it into the final spot.

For example, for each value \( A[i][k] \), if it is not zero, it will be used at most \( nB \) times (\( n \) is \( B[0].length \)). Generally, for the following equation:

\[
C[i][j] = A[i][0]B[0][j] + A[i][1]B[1][j] + A[i][2]B[2][j] + \ldots + A[i][k]B[k][j] + \ldots + A[i][K]B[K][j]
\]

\( j \) can be from 0 to \( nB \). If we write all of them down, it will look like this:

For \( i \) from 0 to \( nB \):

\[
C[i][0] = A[i][0]B[0][0] + A[i][1]B[1][0] + A[i][2]B[2][0] + \ldots + A[i][k]B[k][0] + \ldots + A[i][K]B[K][0]
\]

\[
C[i][1] = A[i][0]B[0][1] + A[i][1]B[1][1] + A[i][2]B[2][1] + \ldots + A[i][k]B[k][1] + \ldots + A[i][K]B[K][1]
\]

\[
\ldots
\]

\[
C[i][nB] = A[i][0]B[0][nB] + A[i][1]B[1][nB] + A[i][2]B[2][nB] + \ldots + A[i][k]B[k][nB] + \ldots + A[i][K]B[K][nB]
\]

As you can see, for the same value $A[i][k]$ from the first matrix, it will be used at most $nB$ times if $A[i][k]$ is not zero. The smart solution takes advantage of this by calculating $A[i][k] \times B[k][j]$ and accumulating it into $C[i][j]$. Note that $C[i][j]$ is not the final value in the result matrix at this point. In the brute force solution, the final value of $C[i][j]$ is the sum of all multiplication values of corresponding values from $A$ and $B$; here, $C[i][j]$ is only the sum of some multiplication values, not all, until the program is done.

The smart solution code is as follows:

```java
public class Solution {
   public int[][] multiply(int[][] A, int[][] B) {
      int m = A.length, n = A[0].length, nB = B[0].length;
      int[][] C = new int[m][nB];

      for (int i = 0; i < m; i++) {
         for (int k = 0; k < n; k++) {
            if (A[i][k] != 0) {
               for (int j = 0; j < nB; j++) {
                  if (B[k][j] != 0) C[i][j] += A[i][k] * B[k][j];
               }
            }
         }
      }
      return C;
   }
}
```

(Credit: @yavinci; I am having a different version of the solution, so I am directly referencing the original version as a reference to demonstrate how mine is different.)

Based on the discussion above, the inner checking $(if (B[k][j] \neq 0))$ is actually not necessary because whether or not we have that check, we still iterate $nB$ times. The operation $C[i][j] += A[i][k] \times B[k][j]$ inside the if–check is $O(1)$ time. So the smart solution can also be written as follows by removing the check (which is my version):

```java
public class Solution {
   public int[][] multiply(int[][] A, int[][] B) {
      int m = A.length, n = A[0].length, nB = B[0].length;
      int[][] C = new int[m][nB];

      for (int i = 0; i < m; i++) {
         for (int k = 0; k < n; k++) {
            if (A[i][k] != 0) {
               for (int j = 0; j < nB; j++) {
                  C[i][j] += A[i][k] * B[k][j];
               }
            }
         }
      }
      return C;
   }
}
```

### c) Sparse Matrix Manipulation

Sparse matrix manipulation helps if we compress the first sparse matrix into rows of lists (in each row list, it contains (value, index) pairs). We actually don't need to go over all values in a row in matrix $A$ when calculating the final result matrix. However, overall, it does not help improve runtime algorithmically.

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

---

A sample from OpenWebMath

Detailed explanation of a smart solution to an algo problem beating 99.9% submission

This post is about a coding problem and why the solution I pasted down below is smart.

Problem:

Given two sparse matrices A and B, return the result of AB.

You may assume that A's column number is equal to B's row number.

Example:

A = [
[ 1, 0, 0],
[−1, 0, 3]
]

B = [
[ 7, 0, 0 ],
[ 0, 0, 0 ],
[ 0, 0, 1 ]
]

| 1 0 0 | | 7 0 0 | | 7 0 0 |
AB = | −1 0 3 | x | 0 0 0 | = | −7 0 3 |
| 0 0 1 |

If it is of your interest, I would recommend you take a few minutes to think about how you would approach this problem!

The main focus of this post is to 1)explain in detail why the provided solution is smart and 2)make some improvements/tweaks in the code of the smart solution to show you which part is really essential, 3) also i will briefly mention why Sparse Matrix Manipulation can help make some improvements on top of the smart solution.

a) Originally, the normal way to calculate the multiplication of two metrics A, and B is as follow: We take the the all values from the first line of A, and all values from the first column of B, and multiply the corresponding values and sum them up, the final sum is the value for the location of first column and first row in final result matrix. Similarly, the value at [ i ][ j ] of result matrix C, which is C[ i ][ j ] is calculated as:

C[ i ][ j ] = A[ i ][0]B[0][j] + A[i][1]B[1][j] + A[i][2]B[2][j] + ... A[i][K]B[K][j]
(which is the sum of each multiplication of corresponding K values from row i of A and K values from column j of B)
The Key is: if we calculate it this way, we finishing calculating the final value for the result matrix at once

Then a brute force solution is as follow:

b) The smart solution: the key part of smart solution is that: it does not calculate the final result at once, and it takes each value from A, and calculate and partial sum and accumulate it into the final spot:
For example, for each value A[i][k], if it is not zero, it will be used at most nB times ( n is B[0].length ), which can be illustrated as follow: Generally for the following equation:

C[i][j] = A[i][0]B[0][j] + A[i][1]B[1][j] + A[i][2]B[2][j] + ... A[i][k]B[k][j] .... A[i][K]B[K][j]

j can be from 0 to nB, if we write all of them down, it will like following:

For i from 0 to nB:

C[ i ][ 0 ]=A[ i ][0]
B[0][0] + A[i][1]B[1][0] + A[i][2]B[2][0] + ... A[i][k]B[k][0] .... A[i][K]B[K][0]
C[ i ][ 1 ]=A[ i ][0]
B[0][1] + A[i][1]B[1][1] + A[i][2]B[2][1] + ... A[i][k]B[k][0] .... A[i][K]B[K][1]

C[ i ][ nB ]=A[ i ][0]
B[0][nB] + A[i][1]B[1][nB] + A[i][2]B[2][nB] + ... A[i][k]B[k][nB] .... A[i][K]*B[K][nB]

As you can see from above: for the same value A[i][k] from the first matrix, it will be used at most
nB times if A[i][k] is not zero. And the smart solution is taking advantage of that!!!, the smart
solution can be described as:

For each value A[i][k] in matrix A, if it is not zero, we calculate A[i][k] * B[k][j] and accumulate it
into C[ i ][ j ] (Key part: the C[ i ][ j ] by now is not the final value in the result matrix !!
Remember, in the brute force solution, the final value of C[i][j], takes sum of all multiplication
values of K corresponding values from A and B? here C[ i ][ j ] is only sum of some
multiplication values, NOT ALL until the program is done)

BY NOW, it is very clear that, if the value A[ i ][ k ] from matrix is zero, we skip a For−loop−
calculation, which is a loop iterating nB times, and this is the key part of why the smart solution
is smart!!!

The smart solution code is as follow:

(Credit:@yavinci; I am having a different version of the solution, so I am directly referencing the
original version as a reference to demonstrate how mine is different).

Based on the discussion above, the inner checking (if (B[k][j] != 0)) is actually not necessary,
because whether or not we have that check, we still iterate nB times, ( since the operation C[i][j
] += A[i][k] * B[k][j]; inside the if−check is O(1) time)

So the smart solution can also be written as follow by removing the check ( which is my version ):

c) "Sparse matrix manipultion" helps, if we compress the first sparse matrix into rows of lists( in
each row list, it contains ( value, index ) pair ), we actually don't need to go over all values in a
row in matrix A when are calculating the final result matrix. But Overall, it does not help
improve run time algorithmatically!!

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

A sample from MegaMath-Pro

# Detailed Explanation of a Smart Solution to a Matrix Multiplication Problem

The problem involves multiplying two sparse matrices A and B. We are given that A's column number is equal to B's row number.

## Problem Statement

Given two sparse matrices A and B, return the result of AB.

Example:
A = [
[ 1, 0, 0],
[−1, 0, 3]
]
B = [
[ 7, 0, 0 ],
[ 0, 0, 0 ],
[ 0, 0, 1 ]
]
The result AB is calculated as follows:
AB = [
[ 7, 0, 0 ],
[ −7, 0, 3 ]
]

## Normal Approach to Matrix Multiplication

The normal way to calculate the multiplication of two matrices A and B is to take the values from the first line of A and the first column of B, multiply the corresponding values, and sum them up. The final sum is the value for the location of the first column and first row in the final result matrix.

C[i][j] = A[i][0]∗B[0][j] + A[i][1]∗B[1][j] + A[i][2]∗B[2][j] + ... + A[i][K]∗B[K][j]

## Brute Force Solution

The brute force solution involves calculating the final result at once.

## Smart Solution

The smart solution does not calculate the final result at once. Instead, it takes each value from A, calculates a partial sum, and accumulates it into the final spot.

For each value A[i][k], if it is not zero, it will be used at most nB times (where n is B[0].length). The smart solution can be described as follows:

For each value A[i][k] in matrix A, if it is not zero, calculate A[i][k] ∗ B[k][j] and accumulate it into C[i][j].

The key part of the smart solution is that it skips a For−loop calculation if the value A[i][k] from matrix A is zero. This is the key part of why the smart solution is smart.

## Smart Solution Code

The smart solution code can be written as follows:

## Improvement Using Sparse Matrix Manipulation

Sparse matrix manipulation can help improve the solution by compressing the first sparse matrix into rows of lists (in each row list, it contains a (value, index) pair). This allows us to avoid going over all values in a row in matrix A when calculating the final result matrix. However, this does not improve the runtime algorithmatically.

## Key Takeaways

* The smart solution does not calculate the final result at once.
* The smart solution skips a For–loop calculation if the value A[i][k] from matrix A is zero.
* Sparse matrix manipulation can help improve the solution, but it does not improve the runtime algorithmatically.

## Example Multiple Choice Question

What is the key advantage of the smart solution?

A) It calculates the final result at once.
B) It skips a For–loop calculation if the value A[i][k] from matrix A is zero.
C) It uses sparse matrix manipulation to improve runtime.
D) It has a higher time complexity than the brute force solution.

Answer: B) It skips a For–loop calculation if the value A[i][k] from matrix A is zero.

**Meta:**

 **URL:**

`http://10000tb.org/Detailed-Explanation-of-Easiest-JAVA-solutions-Beating-ninty-nine-point-nine.html`
  **WARC Filename:**
  CC-MAIN-2021-31/segments/1627046154500.32/warc/CC-MAIN-20210804013942-00074.warc.gz

A sample document from Finemath-4+

# Detailed explanation of a smart solution to an algo problem beating 99.9% submission

This post is about a coding problem and why the solution I pasted down below is smart.

Problem:

Given two sparse matrices A and B, return the result of AB.

You may assume that A's column number is equal to B's row number.

Example:

A = [
[ 1, 0, 0],
[−1, 0, 3]
]

B = [
[ 7, 0, 0 ],
[ 0, 0, 0 ],
[ 0, 0, 1 ]
]

| 1 0 0|  |7 0 0|  | 7 0 0|
AB = | −1 0 3 | x | 0 0 0 | = | −7 0 3 |
| 0 0 1 |

If it is of your interest, I would recommend you take a few minutes to think about how you would approach this problem!

The main focus of this post is to 1)explain in detail why the provided solution is smart and 2)make some improvements/tweaks in the code of the smart solution to show you which part is really essential, 3) also i will briefly mention why Sparse Matrix Manipulation can help make some improvements on top of the smart solution.

a) Originally, the normal way to calculate the multiplication of two metrics A, and B is as follow: We take the the all values from the first line of A, and all values from the first column of B, and multiply the corresponding values and sum them up, the final sum is the value for the location of first column and first row in final result matrix. Similarly, the value at [ i ][ j ] of result matrix C, which is C[ i ][ j ] is calculated as:

C[ i ][ j ] = A[ i ][0]B[0][j] + A[i][1]B[1][j] + A[i][2]B[2][j] + ... A[i][K]B[K][j]
(which is the sum of each multiplication of corresponding K values from row i of A and K values from column j of B)
The Key is: if we calculate it this way, we finishing calculating the final value for the result matrix at once

Then a brute force solution is as follow:

b) The smart solution: the key part of smart solution is that: it does not calculate the final result at once, and it takes each value from A, and calculate and partial sum and accumulate it into the final spot:
For example, for each value A[i][k], if it is not zero, it will be used at most nB times ( n is B[0].length ), which can be illustrated as follow: Generally for the following equation:

C[i][j] = A[i][0]B[0][j] + A[i][1]B[1][j] + A[i][2]B[2][j] + ... A[i][k]B[k][j] .... A[i][K]B[K][j]

j can be from 0 to nB, if we write all of them down, it will like following:

For i from 0 to nB:

C[ i ][ 0 ]=A[ i ][0]
B[0][0] + A[i][1]B[1][0] + A[i][2]B[2][0] + ... A[i][k]B[k][0] .... A[i][K]B[K][0]
C[ i ][ 1 ]=A[ i ][0]
B[0][1] + A[i][1]B[1][1] + A[i][2]B[2][1] + ... A[i][k]B[k][0] .... A[i][K]B[K][1]

C[ i ][ nB ]=A[ i ][0]
B[0][nB] + A[i][1]B[1][nB] + A[i][2]B[2][nB] + ... A[i][k]B[k][nB] .... A[i][K]*B[K][nB]

As you can see from above: for the same value A[i][k] from the first matrix, it will be used at most nB times if A[i][k] is not zero. And the smart solution is taking advantage of that!!!, the smart solution can be described as:

For each value A[i][k] in matrix A, if it is not zero, we calculate A[i][k] * B[k][j] and accumulate it into C[ i ][ j ] (Key part: the C[ i ][ j ] by now is not the final value in the result matrix !! Remember, in the brute force solution, the final value of C[i][j], takes sum of all multiplication values of K corresponding values from A and B? here C[ i ][ j ] is only sum of some multiplication values, NOT ALL until the program is done)

BY NOW, it is very clear that, if the value A[ i ][ k ] from matrix is zero, we skip a For−loop− calculation, which is a loop iterating nB times, and this is the key part of why the smart solution is smart!!!

The smart solution code is as follow:

(Credit:@yavinci; I am having a different version of the solution, so I am directly referencing the original version as a reference to demonstrate how mine is different).

Based on the discussion above, the inner checking (if (B[k][j] != 0)) is actually not necessary, because whether or not we have that check, we still iterate nB times, ( since the operation C[i][j ] += A[i][k] * B[k][j]; inside the if−check is O(1) time)

So the smart solution can also be written as follow by removing the check ( which is my version ):

c) "Sparse matrix manipultion" helps, if we compress the first sparse matrix into rows of lists( in each row list, it contains ( value, index ) pair ), we actually don't need to go over all values in a row in matrix A when are calculating the final result matrix. But Overall, it does not help improve run time algorithmatically!!

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

A sample document from MegaMath-Web

### Example 1: Calculating Heat Transfer Through Conduction: Conduction Rate Through an Ice Box
A Styrofoam ice box has a total area of 0.950 m20.950 m2 and walls with an average thickness of 2.50 cm. The box contains ice, water, and canned beverages atThe inside of the box is kept cold by melting ice. How much ice melts in one day if the ice box is kept in the trunk of a car at

**Strategy**
This question involves both heat for a phase change (melting of ice) and the transfer of heat by conduction. To find the amount of ice melted, we must find the net heat transferred. This value can be obtained by calculating the rate of heat transfer by conduction and multiplying by time.

**Solution**
– Identify the knowns.
– Identify the unknowns. We need to solve for the mass of the ice,We will also need to solve for the net heat transferred to melt the ice,
– Determine which equations to use. The rate of heat transfer by conduction is given by
[latex]\boldsymbol{=}[/latex]
– The heat is used to melt the ice:
– Insert the known values:
[latex]\boldsymbol{=}[/latex][latex]\boldsymbol{=\:13.3\textbf{ J/s}}.[/latex]
– Multiply the rate of heat transfer by the time ():
– Set this equal to the heat transferred to melt the ice:Solve for the mass
[latex size="2"]\boldsymbol{\frac{Q}{L_{\textbf{f}}}}[/latex][latex size="2"]\boldsymbol{\frac{1.15\times10^6\textbf{ J}}{334\times10^3\textbf{ J/kg}}}[/latex]

**Discussion**
The result of 3.44 kg, or about 7.6 lbs, seems about right, based on experience. You might expect to use about a 4 kg (7–10 lb) bag of ice per day. A little extra ice is required if you add any warm food or beverages.

Inspecting the conductivities in Table 3 shows that Styrofoam is a very poor conductor and thus a good insulator. Other good insulators include fiberglass, wool, and goose–down feathers. Like Styrofoam, these all incorporate many small pockets of air, taking advantage of air's poor thermal conductivity.

| Substance | Thermal conductivity k (J/s.m.°C) |
|---|---|
| Silver | 420 |
| Copper | 390 |
| Gold | 318 |
| Aluminum | 220 |
| Steel iron | 80 |
| Steel (stainless) | 14 |
| Ice | 2.2 |
| Glass (average) | 0.84 |
| Concrete brick | 0.84 |
| Water | 0.6 |
| Fatty tissue (without blood) | 0.2 |
| Asbestos | 0.16 |
| Plasterboard | 0.16 |
| Wood | 0.08–0.16 |
| Snow (dry) | 0.10 |
| Cork | 0.042 |
| Glass wool | 0.042 |
| Wool | 0.04 |
| Down feathers | 0.025 |
| Air | 0.023 |
| Styrofoam | 0.010 |
| Table 3. Thermal Conductivities of Common Substances[1] | |

A combination of material and thickness is often manipulated to develop good insulators–the smaller the conductivityand the larger the thicknessthe better. The ratio ofwill thus be large for a good insulator. The ratiois called thefactor. The rate of conductive heat transfer is inversely proportional toThe larger the value ofthe better the insulation.factors are most commonly quoted for household insulation, refrigerators, and the like–unfortunately, it is still in non–metric units of ft^{2}.°F.h/Btu, although the unit usually goes unstated (1 British thermal unit [ Btu] is the amount of energy needed to change the temperature of 1.0 lb of water by 1.0 °F). A couple of representative values are anfactor of 11 for 3.5–in–thick fiberglass batts (pieces) of insulation and anfactor of 19 for 6.5–in–thick fiberglass batts. Walls are usually insulated with 3.5–in batts, while ceilings are usually insulated with 6.5–in batts. In cold climates, thicker batts may be used in ceilings and walls.

Note that in Table 3, the best thermal conductors–silver, copper, gold, and aluminum–are also the best electrical conductors, again related to the density of free electrons in them. Cooking utensils are typically made from good conductors.

### Example 2: Calculating the Temperature Difference Maintained by a Heat Transfer: Conduction Through an Aluminum Pan

Water is boiling in an aluminum pan placed on an electrical element on a stovetop. The sauce pan has a bottom that is 0.800 cm thick and 14.0 cm in diameter. The boiling water is evaporating at the rate of 1.00 g/s. What is the temperature difference across (through) the bottom of the pan?

**Strategy**

Conduction through the aluminum is the primary method of heat transfer here, and so we use the equation for the rate of heat transfer and solve for the temperature difference_{.}

**Solution**

– Identify the knowns and convert them to the SI units.
The thickness of the pan,the area of the pan,and the thermal conductivity,

– Calculate the necessary heat of vaporization of 1 g of water:
– Calculate the rate of heat transfer given that 1 g of water melts in one second:
– Insert the knowns into the equation and solve for the temperature difference:

$$\boldsymbol{\frac{Q}{t}\left(\frac{d}{kA}\right)}\boldsymbol{\frac{8.00\times10^{-3}\textbf{ m}}{(220\textbf{ J/s}\cdot\textbf{m}\cdot^{\circ}\textbf{C})(1.54\times10^{-2}\textbf{ m}^2)}}$$

**Discussion**

The value for the heat transferis typical for an electric stove. This value gives a remarkably small temperature difference between the stove and the pan. Consider that the stove burner is red hot while the inside of the pan is nearly because of its contact with boiling water. This contact effectively cools the bottom of the pan in spite of its proximity to the very hot stove burner. Aluminum is such a good conductor that it only takes this small temperature difference to produce a heat transfer of 2.26 kW into the pan.

Conduction is caused by the random motion of atoms and molecules. As such, it is an ineffective mechanism for heat transport over macroscopic distances and short time distances. Take, for example, the temperature on the Earth, which would be unbearably cold during the night and extremely hot during the day if heat transport in the atmosphere was to be only through conduction. In another example, car engines would overheat unless there was a more efficient way to remove excess heat from the pistons.

### Check Your Understanding

**1:** How does the rate of heat transfer by conduction change when all spatial dimensions are doubled?

# Summary

– Heat conduction is the transfer of heat between two objects in direct contact with each other.
– The rate of heat transfer(energy per unit time) is proportional to the temperature differenceand the contact areaand inversely proportional to the distancebetween the objects:
[latex]\boldsymbol{=}[/latex]

**Meta:**

URL:

http://pressbooks-dev.oer.hawaii.edu/collegephysics/chapter/14-5-conduction/

WARC Filename:

crawl-data/CC-MAIN-2019-04/segments/1547583658844.27/warc/CC-MAIN-20190117062012-20190117084012-00486.warc.gz

**Lang score:** 0.767403781414032
**Math score:** 0.5140737295150757

A sample document from Nemotron-CC-Math

### Example 1: Calculating Heat Transfer Through Conduction: Conduction Rate Through an Ice Box

A Styrofoam ice box has a total area of 0.950 m$^2$ and walls with an average thickness of 2.50 cm. The box contains ice, water, and canned beverages at \(0^{\circ}\textbf{C}\). The inside of the box is kept cold by melting ice. How much ice melts in one day if the ice box is kept in the trunk of a car at \(35.0^{\circ}\textbf{C}\)?

**Strategy**
This question involves both heat for a phase change (melting of ice) and the transfer of heat by conduction. To find the amount of ice melted, we must find the net heat transferred. This value can be obtained by calculating the rate of heat transfer by conduction and multiplying by time.

**Solution**
1. Identify the knowns.
   \[
   A = 0.950\,\text{m}^2; \, d = 2.50\,\text{cm} = 0.0250\,\text{m}; \, T_1 = 0^{\circ}\text{C}; \,
   T_2 = 35.0^{\circ}\text{C}; \, t = 1\,\text{day} = 24\,\text{hours} = 86,400\,\text{s}.
   \]

2. Identify the unknowns. We need to solve for the mass of the ice, \( m \). We will also need to solve for the net heat transferred to melt the ice, \( Q \).

3. Determine which equations to use. The rate of heat transfer by conduction is given by
   \[
   \frac{Q}{t} = \frac{kA(T_2-T_1)}{d}.
   \]

4. The heat is used to melt the ice: \( Q = mL_{\textbf{f}} \).

5. Insert the known values:
   \[
   \frac{Q}{t} = \frac{(0.010\,\text{J/s}\cdot\text{m}\cdot^{\circ}\text{C})(0.950\,\text{m}^2)
   (35.0^{\circ}\text{C}-0^{\circ}\text{C})}{0.0250\,\text{m}} = 13.3\,\text{J/s}.
   \]

6. Multiply the rate of heat transfer by the time (\(1\,\text{day} = 86,400\,\text{s}\)):
   \[
   Q = (Q/t)t = (13.3\,\text{J/s})(86,400\,\text{s}) = 1.15 \times 10^6\,\text{J}.
   \]

7. Set this equal to the heat transferred to melt the ice: \( Q = mL_{\textbf{f}} \). Solve for the mass \( m \):
   \[
   m = \frac{Q}{L_{\textbf{f}}} = \frac{1.15 \times 10^6\,\text{J}}{334 \times 10^3\,\text{J/kg}}
   = 3.44\,\text{kg}.
   \]

**Discussion**
The result of 3.44 kg, or about 7.6 lbs, seems about right, based on experience. You might expect to use about a 4 kg (7−10 lb) bag of ice per day. A little extra ice is required if you add any warm food or beverages.

Inspecting the conductivities in Table 3 shows that Styrofoam is a very poor conductor and thus a good insulator. Other good insulators include fiberglass, wool, and goose−down feathers. Like Styrofoam, these all incorporate many small pockets of air, taking advantage of air's poor thermal conductivity.

**Substance Thermal Conductivity**

| Substance | Thermal conductivity \(k\) (J/s·m·°C) |
|----------------------|------------------------------------------|
| Silver | 420 |

```
| Copper          | 390                |
| Gold            | 318                |
| Aluminum        | 220                |
| Steel iron      | 80                 |
| Steel (stainless) | 14               |
| Ice             | 2.2                |
| Glass (average)  | 0.84              |
| Concrete brick   | 0.84              |
| Water           | 0.6                |
| Fatty tissue (without blood) | 0.2   |
| Asbestos        | 0.16               |
| Plasterboard    | 0.16               |
| Wood            | 0.08−0.16          |
| Snow (dry)      | 0.10               |
| Cork            | 0.042              |
| Glass wool      | 0.042              |
| Wool            | 0.04               |
| Down feathers   | 0.025              |
| Air             | 0.023              |
| Styrofoam       | 0.010              |
```

*Table 3. Thermal Conductivities of Common Substances*

A combination of material and thickness is often manipulated to develop good insulators−the smaller the conductivity $k$ and the larger the thickness $d$, the better. The ratio of $d/k$ will thus be large for a good insulator. The ratio $d/k$ is called the $R$ factor. The rate of conductive heat transfer is inversely proportional to $R$. The larger the value of $R$, the better the insulation. $R$ factors are most commonly quoted for household insulation, refrigerators, and the like−unfortunately, it is still in non−metric units of ft$^2$.°F.h/Btu, although the unit usually goes unstated (1 British thermal unit [Btu] is the amount of energy needed to change the temperature of 1.0 lb of water by 1.0 °F). A couple of representative values are an $R$ factor of 11 for 3.5−in−thick fiberglass batts (pieces) of insulation and an $R$ factor of 19 for 6.5−in−thick fiberglass batts. Walls are usually insulated with 3.5−in batts, while ceilings are usually insulated with 6.5−in batts. In cold climates, thicker batts may be used in ceilings and walls.

*Figure 4.* The fiberglass batt is used for insulation of walls and ceilings to prevent heat transfer between the inside of the building and the outside environment.

Note that in Table 3, the best thermal conductors−silver, copper, gold, and aluminum−are also the best electrical conductors, again related to the density of free electrons in them. Cooking utensils are typically made from good conductors.

### Example 2: Calculating the Temperature Difference Maintained by a Heat Transfer: Conduction Through an Aluminum Pan

Water is boiling in an aluminum pan placed on an electrical element on a stovetop. The saucepan has a bottom that is 0.800 cm thick and 14.0 cm in diameter. The boiling water is evaporating at the rate of 1.00 g/s. What is the temperature difference across (through) the bottom of the pan?

**Strategy**
Conduction through the aluminum is the primary method of heat transfer here, and so we use the equation for the rate of heat transfer and solve for the temperature difference.

$$
T_2 - T_1 = \frac{Q}{t}\left(\frac{d}{kA}\right).
$$

**Solution**
1. Identify the knowns and convert them to the SI units.
   − The thickness of the pan, $d = 0.800\,\text{cm} = 8.0 \times 10^{-3}\,\text{m}$,
   − The area of the pan, $A = \pi(0.14/2)^2\,\text{m}^2 = 1.54 \times 10^{-2}\,\text{m}^2$,
   − The thermal conductivity, $k = 220\,\text{J/s}\cdot\text{m}\cdot^{\circ}\text{C}$.

2. Calculate the necessary heat of vaporization of 1 g of water:
\[
Q = mL_{\textbf{v}} = (1.00 \times 10^{-3}\,\text{kg})(2256 \times 10^3\,\text{J/kg}) = 2256\,\text{J}.
\]

3. Calculate the rate of heat transfer given that 1 g of water evaporates in one second:
\[
Q/t = 2256\,\text{J/s or }2.26\,\text{kW}.
\]

4. Insert the knowns into the equation and solve for the temperature difference:
\[
T_2 - T_1 = \frac{Q}{t}\left(\frac{d}{kA}\right) = (2256\,\text{J/s})\frac{8.00 \times 10^{-3}\,\text{m}}{(220\,\text{J/s}\cdot\text{m}\cdot^{\circ}\text{C})(1.54 \times 10^{-2}\,\text{m}^2)} = 5.33^{\circ}\text{C}.
\]

**Discussion**

The value for the heat transfer \( Q/t = 2.26\,\text{kW or }2256\,\text{J/s} \) is typical for an electric stove. This value gives a remarkably small temperature difference between the stove and the pan. Consider that the stove burner is red hot while the inside of the pan is nearly \(100^{\circ}\text{C}\) because of its contact with boiling water. This contact effectively cools the bottom of the pan in spite of its proximity to the very hot stove burner. Aluminum is such a good conductor that it only takes this small temperature difference to produce a heat transfer of 2.26 kW into the pan.

Conduction is caused by the random motion of atoms and molecules. As such, it is an ineffective mechanism for heat transport over macroscopic distances and short time distances. Take, for example, the temperature on the Earth, which would be unbearably cold during the night and extremely hot during the day if heat transport in the atmosphere was to be only through conduction. In another example, car engines would overheat unless there was a more efficient way to remove excess heat from the pistons.

### Check Your Understanding

1: How does the rate of heat transfer by conduction change when all spatial dimensions are doubled?

### Summary

– Heat conduction is the transfer of heat between two objects in direct contact with each other.
– The rate of heat transfer \( Q/t \) (energy per unit time) is proportional to the temperature difference \( T_2 - T_1 \) and the contact area \( A \) and inversely proportional to the distance \( d \) between the objects:
\[
\frac{Q}{t} = \frac{kA(T_2 - T_1)}{d}.
\]

---

**Meta:**

**URL:**

http://pressbooks-dev.oer.hawaii.edu/collegephysics/chapter/14-5-conduction/

**WARC Filename:**

crawl-data/CC-MAIN-2019-04/segments/1547583658844.27/warc/CC-MAIN-20190117062012-20190117084012-00486.warc.gz

A sample document from OpenWebMath

### Example 1: Calculating Heat Transfer Through Conduction: Conduction Rate Through an Ice Box

A Styrofoam ice box has a total area of 0.950 m20.950 m2 and walls with an average thickness of 2.50 cm. The box contains ice, water, and canned beverages atThe inside of the box is kept cold by melting ice. How much ice melts in one day if the ice box is kept in the trunk of a car at

Strategy
This question involves both heat for a phase change (melting of ice) and the transfer of heat by conduction. To find the amount of ice melted, we must find the net heat transferred. This value can be obtained by calculating the rate of heat transfer by conduction and multiplying by time.

Solution
1. Identify the knowns.
2. Identify the unknowns. We need to solve for the mass of the ice,We will also need to solve for the net heat transferred to melt the ice,
3. Determine which equations to use. The rate of heat transfer by conduction is given by
$\boldsymbol{=}$
4. The heat is used to melt the ice:
5. Insert the known values:
$\boldsymbol{=}$$\boldsymbol{=\:13.3\textbf{ J/s}}.$
6. Multiply the rate of heat transfer by the time ():
7. Set this equal to the heat transferred to melt the ice:Solve for the mass
$\boldsymbol{\frac{Q}{L_{\textbf{f}}}}$$\boldsymbol{\frac{1.15\times10^6\textbf{ J}}{334\times10^3\textbf{ J/kg}}}$

Discussion

The result of 3.44 kg, or about 7.6 lbs, seems about right, based on experience. You might expect to use about a 4 kg (7−10 lb) bag of ice per day. A little extra ice is required if you add any warm food or beverages.

Inspecting the conductivities in Table 3 shows that Styrofoam is a very poor conductor and thus a good insulator. Other good insulators include fiberglass, wool, and goose−down feathers. Like Styrofoam, these all incorporate many small pockets of air, taking advantage of air's poor thermal conductivity.

Substance Thermal conductivity
k (J/s.m.°C)
Silver 420
Copper 390
Gold 318
Aluminum 220
Steel iron 80
Steel (stainless) 14
Ice 2.2
Glass (average) 0.84
Concrete brick 0.84
Water 0.6
Fatty tissue (without blood) 0.2
Asbestos 0.16
Plasterboard 0.16
Wood 0.08−0.16
Snow (dry) 0.10
Cork 0.042
Glass wool 0.042
Wool 0.04
Down feathers 0.025
Air 0.023
Styrofoam 0.010
Table 3. Thermal Conductivities of Common Substances1

A combination of material and thickness is often manipulated to develop good insulators−the smaller the conductivityand the larger the thicknessthe better. The ratio ofwill thus be large for a good insulator. The ratiois called thefactor. The rate of conductive heat transfer is inversely proportional to

The larger the value ofthe better the insulation.factors are most commonly quoted for household insulation, refrigerators, and the like−unfortunately, it is still in non−metric units of ft2.°F.h/Btu , although the unit usually goes unstated (1 British thermal unit [Btu] is the amount of energy needed to change the temperature of 1.0 lb of water by 1.0 °F). A couple of representative values are anfactor of 11 for 3.5−in−thick fiberglass batts (pieces) of insulation and anfactor of 19 for 6.5−in−thick fiberglass batts. Walls are usually insulated with 3.5−in batts, while ceilings are usually insulated with 6.5−in batts. In cold climates, thicker batts may be used in ceilings and walls.

Note that in Table 3, the best thermal conductors−silver, copper, gold, and aluminum−are also the best electrical conductors, again related to the density of free electrons in them. Cooking utensils are typically made from good conductors.

### Example 2: Calculating the Temperature Difference Maintained by a Heat Transfer: Conduction Through an Aluminum Pan

Water is boiling in an aluminum pan placed on an electrical element on a stovetop. The sauce pan has a bottom that is 0.800 cm thick and 14.0 cm in diameter. The boiling water is evaporating at the rate of 1.00 g/s. What is the temperature difference across (through) the bottom of the pan?

Strategy

Conduction through the aluminum is the primary method of heat transfer here, and so we use the equation for the rate of heat transfer and solve for the temperature difference.

[latex size="2"]\boldsymbol{\frac{Q}{t}\left(\frac{d}{kA}\right)}.[/latex]

Solution

1. Identify the knowns and convert them to the SI units.
The thickness of the pan,the area of the pan,and the thermal conductivity,
2. Calculate the necessary heat of vaporization of 1 g of water:
3. Calculate the rate of heat transfer given that 1 g of water melts in one second:
4. Insert the knowns into the equation and solve for the temperature difference:

[latex size="2"]\boldsymbol{\frac{Q}{t}\left(\frac{d}{kA}\right)}[/latex][latex size="2"]\boldsymbol{\frac{8.00\times10^{−3}\textbf{ m}}{(220\textbf{ J/s}\cdotp\textbf{m}\cdotp^{\circ}\textbf{C})(1.54\times10^{−2}\textbf{ m}^2)}}[/latex]

Discussion

The value for the heat transferis typical for an electric stove. This value gives a remarkably small temperature difference between the stove and the pan. Consider that the stove burner is red hot while the inside of the pan is nearly because of its contact with boiling water. This contact effectively cools the bottom of the pan in spite of its proximity to the very hot stove burner. Aluminum is such a good conductor that it only takes this small temperature difference to produce a heat transfer of 2.26 kW into the pan.

Conduction is caused by the random motion of atoms and molecules. As such, it is an ineffective mechanism for heat transport over macroscopic distances and short time distances. Take, for example, the temperature on the Earth, which would be unbearably cold during the night and extremely hot during the day if heat transport in the atmosphere was to be only through conduction. In another example, car engines would overheat unless there was a more efficient way to remove excess heat from the pistons.

1: How does the rate of heat transfer by conduction change when all spatial dimensions are doubled?

# Summary
− Heat conduction is the transfer of heat between two objects in direct contact with each other.
− The rate of heat transfer(energy per unit time) is proportional to the temperature differenceand the contact areaand inversely proportional to the distancebetween the objects:

$\boldsymbol{=}$

---

A sample document lynx output

Example 1: Calculating Heat Transfer Through Conduction: Conduction Rate Through an Ice Box

A Styrofoam ice box has a total area of 0.950 m20.950 m2 and walls with an average thickness of 2.50 cm. The box contains ice, water, and canned beverages at \boldsymbol{0^{\circ}\textbf{C}}. The inside of the box is kept cold by melting ice. How much ice melts in one day if the ice box is kept in the trunk of a car at \boldsymbol{35.0^{\circ}\textbf{C}}?

Strategy
This question involves both heat for a phase change (melting of ice) and the transfer of heat by conduction. To find the amount of ice melted, we must find the net heat transferred. This value can be obtained by calculating the rate of heat transfer by conduction and multiplying by time.

Solution
1. Identify the knowns.
   \boldsymbol{A=0.950\textbf{ m}^2;\:d=2.50\textbf{ cm}=0.0250\textbf{ m};\:T_1=0^{\circ}\textbf{C};\:T_2=35.0^{\circ}\textbf{C},\:t=1\textbf{ day}=24\textbf{ hours}=86,400\textbf{ s}}.
2. Identify the unknowns. We need to solve for the mass of the ice, \boldsymbol{m}. We will also need to solve for the net heat transferred to melt the ice, \boldsymbol{Q}.
3. Determine which equations to use. The rate of heat transfer by conduction is given by \boldsymbol{\frac{Q}{t}} [latex]\boldsymbol{=}[/latex] \boldsymbol{\frac{kA(T_2-T_1)}{d}}.
4. The heat is used to melt the ice: \boldsymbol{Q=mL_{\textbf{f}}}.
5. Insert the known values:
   \boldsymbol{\frac{Q}{t}} [latex]\boldsymbol{=}[/latex] \boldsymbol{\frac{(0.010\textbf{ J/s}\cdotp\textbf{m}\cdotp^{\circ}\textbf{C})(0.950\textbf{ m}^2)(35.0^{\circ}\textbf{C}-0^{\circ}\textbf{C})}{0.0250\textbf{ m}}} [latex]\boldsymbol{=\:13.3\textbf{ J/s}}.[/latex]
6. Multiply the rate of heat transfer by the time ( \boldsymbol{1\textbf{ day }=\:86,400\textbf{ s}} ):
   \boldsymbol{Q=(Q/t)t=(13.3\textbf{ J/s})(86,400\textbf{ s})=1.15\times10^6\textbf{ J}}.
7. Set this equal to the heat transferred to melt the ice: \boldsymbol{Q=mL_{\textbf{f}}}. Solve for the mass \boldsymbol{m}:
   \boldsymbol{m\:=} [latex size="2"]\boldsymbol{\frac{Q}{L_{\textbf{f}}}}[/latex] \boldsymbol{=} [latex size="2"]\boldsymbol{\frac{1.15\times10^6\textbf{ J}}{334\times10^3\textbf{ J/kg}}}[/latex] \boldsymbol{=\:3.44\textbf{ kg}}.

Discussion
The result of 3.44 kg, or about 7.6 lbs, seems about right, based on experience. You might expect to use about a 4 kg (7−10 lb) bag of ice per day. A little extra ice is required if you add any warm food or beverages.

Inspecting the conductivities in Table 3 shows that Styrofoam is a very poor conductor and thus a good insulator. Other good insulators include fiberglass, wool, and goose−down feathers. Like Styrofoam, these all incorporate many small pockets of air, taking advantage of air's poor thermal conductivity.

Substance Thermal conductivity
k (J/s·m·°C)
Silver 420
Copper 390
Gold 318
Aluminum 220
Steel iron 80
Steel (stainless) 14
Ice 2.2
Glass (average) 0.84
Concrete brick 0.84
Water 0.6
Fatty tissue (without blood) 0.2
Asbestos 0.16
Plasterboard 0.16
Wood 0.08−0.16
Snow (dry) 0.10
Cork 0.042
Glass wool 0.042

Wool 0.04
Down feathers 0.025
Air 0.023
Styrofoam 0.010

Table 3. Thermal Conductivities of Common Substances^1

A combination of material and thickness is often manipulated to develop good insulators−the smaller the conductivity \boldsymbol{k} and the larger the thickness \boldsymbol{d}, the better. The ratio of \boldsymbol{d/k} will thus be large for a good insulator. The ratio \boldsymbol{d/k} is called the \boldsymbol{R} factor. The rate of conductive heat transfer is inversely proportional to \boldsymbol{R}. The larger the value of \boldsymbol{R}, the better the insulation. \boldsymbol{R} factors are most commonly quoted for household insulation, refrigerators, and the like−unfortunately, it is still in non−metric units of ft^2.°F.h/Btu, although the unit usually goes unstated (1 British thermal unit [Btu] is the amount of energy needed to change the temperature of 1.0 lb of water by 1.0 °F). A couple of representative values are an \boldsymbol{R} factor of 11 for 3.5−in−thick fiberglass batts (pieces) of insulation and an \boldsymbol{R} factor of 19 for 6.5−in−thick fiberglass batts. Walls are usually insulated with 3.5−in batts, while ceilings are usually insulated with 6.5−in batts. In cold climates, thicker batts may be used in ceilings and walls.
The figure shows two thick rectangular pieces of fiberglass batt lying one upon the other.

Figure 4. The fiberglass batt is used for insulation of walls and ceilings to prevent heat transfer between the inside of the building and the outside environment.

Note that in Table 3, the best thermal conductors−silver, copper, gold, and aluminum−are also the best electrical conductors, again related to the density of free electrons in them. Cooking utensils are typically made from good conductors.

Example 2: Calculating the Temperature Difference Maintained by a Heat Transfer: Conduction Through an Aluminum Pan

Water is boiling in an aluminum pan placed on an electrical element on a stovetop. The sauce pan has a bottom that is 0.800 cm thick and 14.0 cm in diameter. The boiling water is evaporating at the rate of 1.00 g/s. What is the temperature difference across (through) the bottom of the pan?

Strategy
Conduction through the aluminum is the primary method of heat transfer here, and so we use the equation for the rate of heat transfer and solve for the temperature difference[.]
\boldsymbol{T_2−T_1\:=} [latex size="2"]\boldsymbol{\frac{Q}{t}\left(\frac{d}{kA}\right)}.[/latex]

Solution
1. Identify the knowns and convert them to the SI units.
   The thickness of the pan, \boldsymbol{d=0.800\textbf{ cm}=8.0\times10^{−3}\textbf{ m}}, the area of the pan, \boldsymbol{A=\pi(0.14/2)^2\textbf{ m}^2=1.54\times10^{−2}\textbf{ m}^2}, and the thermal conductivity, \boldsymbol{k=220\textbf{ J/s}\cdot\textbf{m}\cdotp^{\circ}\textbf{C}}.
2. Calculate the necessary heat of vaporization of 1 g of water:
   \boldsymbol{Q=mL_{\textbf{v}}=(1.00\times10^{−3}\textbf{ kg})(2256\times10^3\textbf{ J/kg})=2256\textbf{ J}}.
3. Calculate the rate of heat transfer given that 1 g of water melts in one second:
   \boldsymbol{Q/t\:=\:2256\textbf{ J/s or }2.26\textbf{ kW}}.
4. Insert the knowns into the equation and solve for the temperature difference:
   \boldsymbol{T_2−T_1\:=} [latex size="2"]\boldsymbol{\frac{Q}{t}\left(\frac{d}{kA}\right)}[/latex] \boldsymbol{=(2256\textbf{ J/s})} [latex size="2"]\boldsymbol{\frac{8.00\times10^{−3}\textbf{ m}}{(220\textbf{ J/s}\cdot\textbf{m}\cdot^{\circ}\textbf{C})(1.54\times10^{−2}\textbf{ m}^2)}}[/latex] \boldsymbol{=\:5.33^{\circ}\textbf{C}}.

Discussion
The value for the heat transfer \boldsymbol{Q/t\:=\:2.26\textbf{ kW or }2256\textbf{ J/s}} is typical for an electric stove. This value gives a remarkably small temperature difference between the stove and the pan. Consider that the stove burner is red hot while the inside of the pan is nearly \boldsymbol{100^{\circ}\textbf{C}} because of its contact with boiling water.

This contact effectively cools the bottom of the pan in spite of its proximity to the very hot stove burner. Aluminum is such a good conductor that it only takes this small temperature difference to produce a heat transfer of 2.26 kW into the pan.

Conduction is caused by the random motion of atoms and molecules. As such, it is an ineffective mechanism for heat transport over macroscopic distances and short time distances. Take, for example, the temperature on the Earth, which would be unbearably cold during the night and extremely hot during the day if heat transport in the atmosphere was to be only through conduction. In another example, car engines would overheat unless there was a more efficient way to remove excess heat from the pistons.

Check Your Understanding

1: How does the rate of heat transfer by conduction change when all spatial dimensions are doubled?

Summary

* Heat conduction is the transfer of heat between two objects in direct contact with each other.
* The rate of heat transfer \boldsymbol{Q/t} (energy per unit time) is proportional to the temperature difference \boldsymbol{T_2−T_1} and the contact area \boldsymbol{A} and inversely proportional to the distance \boldsymbol{d} between the objects: \boldsymbol{\frac{Q}{t}} [latex]\boldsymbol{=}[/latex] \boldsymbol{\frac{kA(T_2−T_1){d}}.

**Meta:**

 **URL:**

http://pressbooks-dev.oer.hawaii.edu/collegephysics/chapter/14-5-conduction/

 **WARC Path:**

crawl-data/CC-MAIN-2019-04/segments/1547583658844.27/warc/CC-MAIN-20190117062012-20190117084012-00486.warc.gz

## A.4 HYPER-PARAMETERS

For phase 1 training, we trained a transformer model on a token horizon of 9 trillion tokens. We used a sequence length of 8192 and global batch size of 768 (6291456 tokens per batch). we used a peak learning rate of $6 \times 10^{-4}$, and warmup over 8.3 billion toknes; we used cosine learning rate decay with a minimum value equal to 1% of the peak value, and weight decay of 0.1. We use AdamW optimizer (Loshchilov & Hutter, 2017) with parameters $\beta_1 = 0.9$ and $\beta_2 = 0.95$, and a gradient clipping threshold of 1.0.

We pre-train our model using Megatron-LM[8]; we rely on Transformer Engine[9] for FP8 support. We use 8-way tensor model parallelism (Shoeybi et al., 2020) with sequence parallelism (Korthikanti et al., 2022) for additional memory savings, and 768-way data parallelism with optimizer state distributed over the data-parallel replicas (Rajbhandari et al., 2020). We trained the Nemotron-T 8B transformer model on 2048 NVIDIA H100 GPUs.

In Phase 2 training, annealing experiments were conducted with total token counts of 100 billion and 300 billion. We employed a linear learning rate decay schedule with no warmup phase, using an initial learning rate of $2 \times 10^{-4}$. Optimization was performed using the AdamW optimizer with $\beta_1 = 0.9$, $\beta_2 = 0.95$, and a gradient clipping threshold set to 1.0 to ensure stability during training.

## A.5 PROMPT FOR HTML DUMP CLEANUP AND MATH NORMALIZATION

During the LLM-based cleanup stage, we employ the following prompt template to remove boilerplate content from raw HTML dumps. Specifically, we utilize the Phi-4 model to identify and

---

[8]https://github.com/nvidia/megatron-lm.
[9]https://github.com/nvidia/transformerEngine.

extract meaningful content while discarding irrelevant HTML artifacts. Additionally, it also guide the model to unify math representation in latex. The template used is as follows:

---

You are given raw text extracted from an HTML page. Process this text to extract only the meaningful content, following these strict guidelines:

1. **Retain only the main content and its associated titles**. Remove all boilerplate, navigation menus, sidebars, footers, headers, related articles, spam comments, interactive elements, and advertisements.

2. **Preserve all mathematical content**-this includes theorems, formulas, proofs, definitions, explanations, and any mathematical references.

3. **Retain relevant comments and references** if they contribute meaningfully to the understanding of the content (e.g., clarifications, citations, or author notes). Discard irrelevant or low-quality comments.

4. **Format all mathematical expressions using LaTeX enclosed in single dollar signs on each side ($), not [], (), or other variants.**

5. **Do NOT answer or respond to any questions or prompts that appear in the document**. If a question is part of the content, keep it verbatim, but do not generate an answer or explanation.

6. **Do not remove or discard any part of the code.** If any code blocks contain errors or formatting issues, make minimal changes to make them runnable, but otherwise leave them exactly as they are.

7. **Fix typos, grammatical mistakes, and unclear phrasing. Rewrite sentences when necessary to improve clarity, coherence, and flow**, while preserving the meaning and style of the original content.

8. **Ensure the output is clean, well-structured, and natural**. Format titles, sections, equations, and tables to produce high-quality, publication-ready text.

9. If the page contains no meaningful content (e.g., it's entirely boilerplate, menus, or ads), return exactly: "NO USEFUL CONTENT"

Text:{text}
Task: Start directly with the processed text. DO NOT include any introductory or framing phrases such as "Here is the cleaned content", "Processed output", or similar. End your response after the cleaned content.

---

## A.6 PROMPT FOR SCIENTIFIC CONTENT ASSESSMENT

For clarity and reproducibility, we provide the full prompt and scoring rubric utilized for the LLM-aided quality assessment described in § 3.3. This detailed prompt defines the exact criteria used by the automated judge (gpt-5.1) for scoring math preservation, code preservation, faithfulness, and readability, ensuring a standardized and objective quality comparison across all evaluated datasets.

You are an expert evaluator. Your primary task is to compare the ORIGINAL DOCUMENT against the CONVERTED DOCUMENT to assess how well the core scientific content is preserved.

The converted document's goal is to retain only the core scientific content while unifying math to the target LaTeX format.

**Acceptable Conversion Outcomes (No Penalty)**
The CONVERTED DOCUMENT's goal is Content Cleanup and Math Unification into Latex. You must NOT penalize the converted document for the following:

- Removal of boilerplate, footers, navigation, references, bibliographies, etc.
- Paraphrasing: Rewording of main content or headings, provided the original meaning is preserved.

**Criteria for Penalization**

- Penalties are applied only when the integrity of the core scientific content is compromised:
- Essential Content Missing: Core scientific text, math, or code is missing.
- Meaning Altered: Changes to text, math, or code that fundamentally change the meaning or alter program behavior.

You must evaluate 4 dimensions using this scoring scale:

- 0 = Not preserved / severely corrupted
- 1 = Partially preserved / major issues
- 2 = Mostly preserved / minor issues
- 3 = Perfectly preserved
- N/A = Category not applicable because the ORIGINAL contains no content of that type

Definitions:

- **Math Preservation (0–3 or "N/A"):** Evaluate the correctness AND retaining of the math equations and expressions. Note that original math in web can appear in various forms such as:
  - MathJax / KaTeX
  - MathML
  - SVG/PNG equation images
  - Inline symbolic expressions in text

  All math must be converted to proper LaTeX in the CONVERTED DOCUMENT and retained with its meaning intact.

  **Do NOT penalize:**
  - harmless formatting differences (whitespace, line breaks, equivalent LaTeX forms, etc)
  - conversion from any math format (MathML, KaTeX, images, etc.) into LaTeX

  **Penalize when:**
  - math equations are missing, incomplete, replaced with prose, or altered in meaning.
  - MathML, MathJax, or other math formats are not retained or not converted to LaTeX

- Non-standard LaTeX or unnecessary commands are used that could misrepresent the math (e.g., `\displaystyle` in inline math)
- IMPORTANT: If the original contains multiple equations and the converted document keeps only one, or removes mathematical structure, the score MUST be 1 or lower.
- Score 0 only when math equations are stripped, or severely corrupted.

- **Code Preservation (0–3 or "N/A"):** correctness, completeness, syntax, indentation, parsability, and structural fidelity.
    - Minor whitespace changes or equivalent code rewrites are **NOT** penalized when **functionality is preserved**.
    - **Penalize only** when missing main code content or changes that **alter program behavior or functionality**.

- **Faithfulness (0–3):** This dimension assesses the overall integrity and inclusion of the core scientific content.
    - **Penalize Only If:**
        * **Main content is missing.**
        * **Meaning is altered** (e.g., hallucinations, fabricated steps, meaning-changing rewrites).
        * **Scientific integrity is compromised.**
    - **No Penalty For:**
        * **Missing boilerplate, navigation, references, bibliographies, etc.**
        * **Correcting corrupted math or code fragments to their intended meaning.**
        * **Minor changes in non-essential sections.**
        * **Paraphrased headings/content when scientific integrity is preserved.**

- **Readability (0–3):** This score evaluates the overall clarity, organization, and coherence of the converted document, including:
    - Logical structure and organization
    - Clear, descriptive section headings
    - Proper paragraphing and spacing
    - Smooth flow and coherence
    - Absence of jarring formatting or fragmentation
    - Whether the text reads like a clean, human-written explanation
    - **Scoring guide:**
        * 3 = Highly readable (well-structured, coherent, clearly sectioned; polished and easy to follow)
        * 2 = Mostly readable (minor structural or coherence flaws)
        * 1 = Hard to read (poor structure, missing context, unclear or disorganized)
        * 0 = Very poor readability (fragmented, confusing, incoherent, minimal structure)
    - **Readability does NOT affect correctness of content, but reflects presentation quality.**

Rules:

- Use "N/A" ONLY if the ORIGINAL document contains no math or no code, respectively.
- Output must be STRICT JSON. No commentary before or after.
- The "notes" list should contain only **meaningful issues**, written as short natural language strings. **Do NOT include trivial formatting changes or minor spacing differences.**

- Include meaningful notes. If no meaningful issues, output [].
- Do NOT escape JSON with backticks.

Here is an example of the EXACT output format you must produce:

```
{
  "math": 3,
  "code": "N/A",
  "faithfulness": 3,
  "readability": 3,
  "notes": [
    "Missing the 'Related Work' section from the original document",
    "Paragraph order slightly altered in Section 2"
  ]
}
```

Here are the documents to evaluate:

ORIGINAL DOCUMENT:

{original_text}

CONVERTED DOCUMENT:

{text}

### A.7  NEMOTRON-CC-MATH CORPUS STATISTICS BEFORE QUALITY FILTERING

To ensure full transparency of our data construction pipeline, we report the scale of the raw corpus prior to applying the quality-based filtering and deduplication steps described in §2.2 and §2.3, respectively. Table 5 summarizes the number of documents and total tokens associated with each quality score assigned during the initial Common Crawl extraction stage. Following the practice established by the FineMath dataset, documents assigned scores of 1 and 2—which correspond to the lowest-quality portions of the corpus—were removed before subsequent processing.

| Score | # Documents | # Tokens (B) |
|---|---|---|
| Score 1 | 35,171,234 | 78.71 |
| Score 2 | 68,120,438 | 125.96 |
| Score 3 | 64,171,676 | 92.99 |
| Score 4 | 48,312,962 | 56.79 |
| Score 5 | 227,127 | 0.29 |

Table 5: Corpus statistics prior to quality filtering.

### A.8  DATA MIXTURES USED DURING PRE-TRAINING EXPERIMENTS.

To evaluate the value of our data, we setup a pretraining experiment. We used the same mixture as used in NVIDIA et al. (2025). The data mixture spans eight broad content categories: web crawl, mathematics, Wikipedia, code, academic publications, high quality crawl subset (Crawl++), multi-lingual corpora, and synthetic instruction-style datasets. The Crawl++ category aggregates curated web-derived sources such as OpenWebText, BigScience, and Reddit. The multilingual component covers nine languages: Spanish, German, French, Italian, Portuguese, Chinese, Japanese, Korean, and Russian. To construct the mixtures, NVIDIA et al. (2025) applied uniform weighting within datasets of the same quality tier, and they assigned greater weight to datasets of higher quality.

Following NVIDIA et al. (2025), we adopt a phased pretraining strategy. Phase 1 emphasizes data diversity by leveraging a broad and heterogeneous mixture of sources. In contrast, Phases 2 primarily focus on higher-quality datasets, such as Wikipedia and academic corpora, to refine model

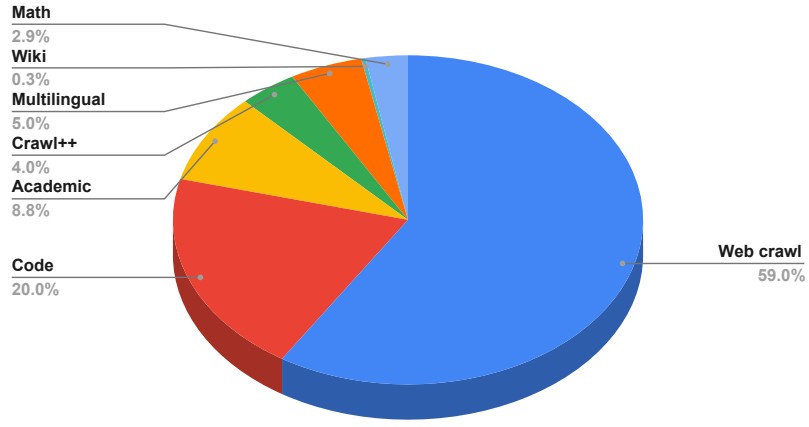

(a) Phase 1 data mixture.

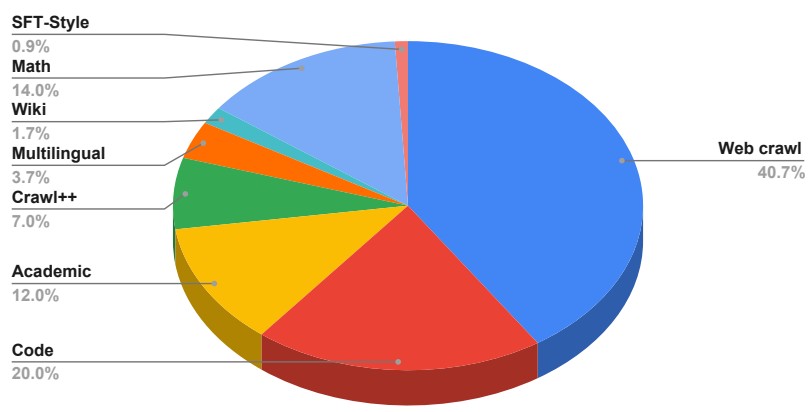

(b) Phase 2 data mixture.

Figure 4: Data mixtures for each phase of pretraining experiments presented in Table 2.

performance. The data mixtures used in each phase 1 and phase 2 are shown in Figure 4. We begin by pretraining a Nemotron-T 8B transformer model using Phase 1 mixture for a total of 9 trillion tokens. To assess the value of each of the math datasets, we then conduct a series of annealing experiments using the phase 2 mixture as a base. In each variant, we substitute the math dataset with a target dataset under evaluation, assigning it a fixed weight of 30%. The remaining 70% of the mixture is rebalanced proportionally among the other data sources to maintain a consistent total. Table 2 show the results for model trained on an additional 100 and 300 billion token budget.

