# OpenReview forum: "Nemotron-CC-Math: A 133 Billion-Token-Scale High Quality Math Pretraining Dataset"
_ICLR.cc/2026/Conference — ICLR 2026 Poster_

### Official Review · Reviewer_AQgW · 2025-10-31

**Soundness:** 4
**Presentation:** 4
**Contribution:** 3
**Rating:** 6
**Confidence:** 5

**Summary:**

This paper redesigns the text extraction pipeline for web data curation, firstly employing Lynx, a text-based browser, to retain texts, and employing Phi-4 to rewrite text for quality, resulting in Nemotron-CC-Math. It delivers better data quality than existing the-state-of-the-art open-source corpora, such as MegaMath, FineMath.

**Strengths:**

1. Well-written and structured paper, solid experiments;
2. The lynx’s introduction, which reliably captures equations and maintains code indentation, avoids the heuristics DOM tree operations, such as MegaMath.
3. The ablation on different refinement models is solid.

**Weaknesses:**

I believe that the effectiveness of Lynx should be evaluated through an apples-to-apples comparison. For example, the quality of Lynx versus DOM tree optimization (as introduced in MegaMath) on the same mathematical web pages could be compared under a controlled setting.

**Questions:**

In addition, the potential negative impact introduced during the document refinement process should be clearly discussed. I believe this could be an important point that warrants further analysis.

---

> ### Author Response · Authors · 2025-11-19
> **Response to Reviewer AQgW**
>
> We thank the reviewer for thoughtful comments and the recognition of Lynx's structural effectiveness. We have implemented a new, rigorous quantitative analysis to directly address your concerns.
>
> >> "I believe that the effectiveness of Lynx should be evaluated through an apples-to-apples comparison."
>
> We agree and have addressed this by adding a new Quality Assessment (Section 3.3). Although the shared document subset is necessarily small (97,788 pages), this quantitative, apples-to-apples comparison validates our approach:
>
> - **Our Lynx-based pipeline achieves substantially higher Math Preservation** (2.55 vs. MegaMath: 1.85, OWM: 2.28, Finemath: 2.31).
> - It achieves **substantially higher Code Preservation** (3.00 vs. MegaMath: 1.75, Finemath/OWM: 2.5), confirming its structural superiority.
>
> >> "In addition, the potential negative impact introduced during the document refinement process should be clearly discussed. I believe this could be an important point that warrants further analysis."
>
> The new **Section 3.3** directly quantifies this trade-off. While our Faithfulness score (2.44) is marginally lower than OWM (2.59) and FineMath (2.63)—the expected result of the LLM-based cleanup and compressing/rewriting text—it remains **substantially higher than MegaMath** (1.92), which also uses LLM refinement. Crucially, our process yields the **highest Readability score** (2.90) among all evaluated datasets (vs. MegaMath: 2.87, OWM: 2.22, FineMath: 2.27).
> This demonstrates that the refinement step successfully trades minimal literal loss for **superior textual coherence** without compromising core structural integrity.
>
> We believe the new Section 3.3 quantitatively resolves your concern, and we respectfully ask you to consider raising your score.

---

### Official Review · Reviewer_yQeP · 2025-11-01

**Soundness:** 3
**Presentation:** 3
**Contribution:** 3
**Rating:** 8
**Confidence:** 4

**Summary:**

This paper introduces Nemotron-CC-Math, a large, high-quality math corpus built from Common Crawl. It uses a domain-agnostic extraction pipeline: layout-aware `lynx` rendering plus structure-preserving LLM cleaning to unify MathJax/KaTeX/MathML into LaTeX, keep equation and code structure, and remove boilerplate.

It releases two datasets: Nemotron-CC-Math-3+ (133B tokens) and Nemotron-CC-Math-4+ (52B tokens), claimed to surpass MegaMath, FineMath, and OpenWebMath; 4+ has about 5.5× the tokens of FineMath-4+. As for experiments, the authors show that pretraining an 8B model on this data yields 4.8-12.6 on MATH and 4.6-14.3 on MBPP+, with additional gains on benchmarks like MMLU.

**Strengths:**

1. I think overall the data pipelines are very sound. the authors combines layout-aware lynx rendering with structure-preserving LLM cleaning, avoiding information loss from naïve HTML-to-text extraction.

2. The authors also unify MathJax/KaTeX/MathML into LaTeX, preserving equation and code structure while removing boilerplate, which is very important but often under-estimated in previous works.

2. The experiments show very promising results, further demonstrating the quality of the datasets.

**Weaknesses:**

I don't see any obvious weaknesses.

**Questions:**

I do have some questions for the authors:

1. why pre-train on math also boosts code performance? if so, have you compared this with other math&code-related datasets? such as stack-edu, megamath-code?

2. How do you detect and constrain LLM over-editing or hallucinations (e.g., symbol renaming, citation mismatches, skipped derivations)? Did you conduct manual sampling review and an inter-annotator agreement (IAA) evaluation?

3. Could you report quantitative results for LaTeX/code parseability, structural consistency (e.g., AST-based edit distance), and rendering consistency?

---

> ### Author Response · Authors · 2025-11-19
> **Response to Reviewer yQeP**
>
> We thank the reviewer for recognizing the soundness of our pipeline, the importance of unifying mathematical structure, and the promising experimental results. Your questions are insightful and have been addressed with a new quantitative analysis.
>
> >> why pre-train on math also boosts code performance? if so, have you compared this with other math&code-related datasets? such as stack-edu, megamath-code?
>
> The boost in code performance is an expected result, as prior work confirms that **code and mathematical reasoning capabilities are highly coupled** (see section 4.2 in Kou et al. (2025) [1], section 5.1.1 in Shao et al. (2024) [2], Lu et al. (2024) [3]). Specifically:
>
> - **Code Retention:** Our Lynx-based pipeline, unlike previous HTML parsers, preserves code snippets in full, including crucial indentation and structure (see section 2.1.1 and Appendix A.3.2). Nemotron-CC-Math-3+ and 4+ contain approximately 4.3M and 1.44M high-quality code samples, respectively, which directly contributes to code skill development (see section 3.1).
> - **LLM Refinement:** Our refinement prompt (Appendix A.5) explicitly instructs the LLM to preserve and correct formatting issues in code, further ensuring quality.
>
> **Our primary focus in this work is addressing the challenges of mathematical content extraction from noisy HTML.** While the observed code performance boost is a valuable **side benefit**, a direct, apples-to-apples comparison with specialized code datasets (like Stack-Edu or MegaMath-Code) is not feasible. These datasets are derived from different source materials (e.g., GitHub pages) and employ code-specific filtering not relevant to our core task of solving HTML extraction challenges for math and incidental code.
>
>
> >> "How do you detect and constrain LLM over-editing or hallucinations (e.g., symbol renaming, citation mismatches, skipped derivations)? Did you conduct a manual sampling review and an inter-annotator agreement (IAA) evaluation?"
>
> We constrained LLM over-editing through a carefully designed, conservative cleanup prompt (Appendix A.5), instructing the model to remove only boilerplate while rigidly preserving all equations, code, and main content.
> Since IAA is not applicable to our case (assessing LLM refinement on unlabeled, large-scale web text), we instead provided rigorous quantitative evidence in the new Section 3.3:
>
> - **Quantitative Proof of Control (Faithfulness):** Our LLM-aided assessment shows our Faithfulness score (2.44) is only marginally lower than non-LLM baselines (expected due to intentional rewrite/compression when using an LLM) but is substantially higher than MegaMath (1.92), which also uses LLM refinement. This confirms our prompt is highly effective at controlling semantic drift and that our Lynx-based pipeline effectively preserves content.
> - **Qualitative & Ablation Verification:** Our ablation study (Section 3.2) shows that Phi-4 achieves consistent cleanup without introducing semantic drift, matching or exceeding the output quality of much larger models. Qualitative examples (Appendix A.3) further verify that the cleanup introduces minimal distortion while improving global coherence. Moreover, downstream gains (e.g., +12.6 on MATH) confirm the quality of our dataset.
>
>
> >> "Could you report quantitative results for LaTeX/code parseability, structural consistency (e.g., AST-based edit distance), and rendering consistency?"
>
> We have directly addressed this concern by adding the Quality Assessment (Section 3.3), which provides quantitative proxies for structural and rendering consistency:
>
> - **Superior Structural Fidelity:** Scored on a 0-3 scale, our pipeline obtains the highest scores for both **Math Preservation** (2.55) and **Code Preservation** (3.00). This confirms that the Lynx pipeline and subsequent refinement successfully ensure high parseability and structural consistency, far exceeding baselines (e.g., MegaMath Code Preservation: 1.75).
>
> - **Highest Text Quality:** Our data also obtains the **highest Readability score** (2.90). This metric confirms the content is readily understandable and well-formatted for rendering.
>
> Given that our new analysis quantitatively demonstrates **superior structural fidelity** and **high text quality** resulting from our pipeline, we believe this fully addresses your question.
>
>
>
>
>
> [1] Which Data Attributes Stimulate Math and Code Reasoning? An Investigation via Influence Functions, Kou et al,  https://arxiv.org/pdf/2505.19949.
>
> [2] DeepSeekMath: Pushing the Limits of Mathematical Reasoning in Open Language Models, Shao et al, 2024.
>
> [3] MathCoder2: Better Math Reasoning from Continued Pretraining on Mathematical Code: This also shows the other way around that code improves math, Lu et al, 2024, https://arxiv.org/pdf/2410.08196.

---

### Official Review · Reviewer_upxT · 2025-11-03

**Soundness:** 4
**Presentation:** 4
**Contribution:** 4
**Rating:** 8
**Confidence:** 4

**Summary:**

This paper introduces nemotron-cc-math, a math dataset created by utilizing the Lynx text-based browser. By using the browser, the HTML can be rendered as structured format with equation and code layout as human read them. The content is then fed through an LLM (Phi-4 14B) to clean up the boilerplates. The final dataset retains data passing Fineweb classifier (3+), which contains 133B tokens, which is one of the largest set at this quality.

**Strengths:**

- Robust and Proper Pipeline: The lynx + LLM-cleaner pipeline is an effective solution. It addresses the failure mode of previous web math extractors that corrupt math and code. The qualitative examples provided clearly demonstrate its superiority in preserving structure. While this method makes sense, both the lynx rendering and an 14B LLM cleaner are expensive for many practitioners. So the open sharing of this resource will help the community significantly.
- The paper delivers a dataset that is both larger and higher quality than existing open-source alternatives. The 133B high quality portion is larger than FineMath, though smaller than the 300B MegaMath.
- Strong experiment results: the methods are tested on an 8B mid-training checkpoint, with 100B and 300B experiments. The scale of the experiment should be sufficient and it shows good results on a range of benchmarks.
- Once again, the contribution to open science and open source of this work is commendable.

**Weaknesses:**

This paper can benefit from additional experimental settings. The main experiments are conducted at the mid-train setting. Will there be some confounding factor from the base model itself? Further, would larger amount of unique tokens be helpful and how much repetition can this dataset be used?

The readers would also benefit from learning about the filtered out portion, i.e., Nemotron-cc-math-1-3. The token size, quality and corresponding model performance may provide valuable information.

**Questions:**

How much total tokens are their without the quality filter? Is there an estimate?

---

> ### Author Response · Authors · 2025-11-19
> **Response to Reviewer upxT**
>
> We appreciate your highly encouraging review and are delighted that you found our contribution to open science commendable.
>
> >> The main experiments are conducted at the mid-train setting. Will there be some confounding factor from the base model itself?
>
> Our use of a mid-train checkpoint for annealing ablations follows best practices from prior literature to estimate the isolated contribution and quality of the math dataset. This approach is necessary because math and code abilities primarily arise only after extensive initial pretraining (see section 3: Blakeney et al. (2024); Dubey et al. (2024); OLMo et al. (2024); Allal et al. (2025)). Moreover, to effectively isolate the math data's contribution within a real-world mixed blend, we upweight the target math dataset—a common practice in continued pretraining literature (see section 4 in Parmar et al. (2024) [1], Gupta et al. (2023) [2] ). This methodology provides a rigorous quality estimate by minimizing confounding factors.
>
> >> Further, would larger amount of unique tokens be helpful and how much repetition can this dataset be used?
>
> This is an excellent question. We have observed that repetition is highly effective for specialization. We have done analysis in our past experiments, where we trained an MoE model from scratch with math data repeated across 1, 2, and 4 epochs (increasing the weight of the math dataset in the blend) shows that math scores continue to improve with repetition.
>
> | Benchmark     | Math 1 epoch | Math 2 epochs | Math 4 epochs |
> |---------------|--------------|---------------|----------------|
> | **Knowledge** |              |               |                |
> | MMLU-Pro      | 14.88        | 16.13         | 15.71          |
> | MMLU          | 51.23        | 49.84         | 50.10          |
> | **Code**      |              |               |                |
> | HumanEval+    | 26.59        | 25.37         | 26.89          |
> | MBPP+         | 34.25        | 34.62         | 35.15          |
> | MBPP          | 44.71        | 43.60         | 44.42          |
> | HumanEval     | 30.98        | 28.23         | 30.27          |
> | **Math**      |              |               |                |
> | Math          | 8.20         | 10.80         | 12.60          |
> | GSM8K         | 24.56        | 28.28         | 35.10          |
>
> Blend information:
>
> | Source         | Blend 1 | Blend 2 | Blend 3 |
> |----------------|---------|---------|---------|
> | crawl          | 70.63%  | 68.97%  | 65.65%  |
> | math           | 1.67%   | 3.33%   | 6.65%   |
> | stack-exchange | 0.40%   | 0.40%   | 0.40%   |
> | wiki           | 0.30%   | 0.30%   | 0.30%   |
> | code           | 20.00%  | 20.00%  | 20.00%  |
> | crawl++        | 4.00%   | 4.00%   | 4.00%   |
> | multilingual   | 3.00%   | 3.00%   | 3.00%   |
>
> We prepare our data into two quality buckets: 4+ (highest quality) and 3+ (mid quality), similar to the FineMath dataset, to account for the limited supply of unique, highest-quality tokens. The 3+ bucket offers greater volume, and our experiments confirm that longer training on 3+ substantially improves scores across math and other benchmarks, making it suitable for users needing a larger token budget (see table 2, also copied here)
>
>
> | Benchmark     | 100B tokens | 300B tokens |
> |---------------|-------------|-------------|
> | **Knowledge** |             |             |
> | MMLU-Pro      | 38.49       | 39.32       |
> | MMLU          | 67.55       | 68.20       |
> | MMLU-Stem     | 62.67       | 64.26       |
> | **Code**      |             |             |
> | HumanEval+    | 34.82       | 37.16       |
> | MBPP+         | 45.11       | 43.51       |
> | MBPP          | 53.48       | 56.15       |
> | HumanEval     | 38.93       | 40.30       |
> | **Math**      |             |             |
> | Math          | 40.60       | 44.20       |
> | GSM8K         | 76.27       | 80.06       |
>
> >> Data statistics of filtered out portion, i.e., Nemotron-cc-math-1-3.
>
> We appreciate this important question. We filtered out scores 1 and 2 following our detailed quality analysis (which revealed low quality and high duplication) and the precedent set by FineMath. We retained the higher-quality data (scores 3-5) for pretraining. The pre-deduplication document and token distribution by score is now detailed in Appendix A.7 of the revised draft, also detailed her:
>
>
> | Score    | # Documents | # Tokens |
> |----------|-------------|----------|
> | Score 1  | 35,171,234  | 78.71B   |
> | Score 2  | 68,120,438  | 125.96B  |
> | Score 3  | 64,171,676  | 92.99B   |
> | Score 4  | 48,312,962  | 56.79B   |
> | Score 5  | 227,127     | 0.29B    |
>
>
>
> [1] Reuse, Don’t Retrain: A Recipe for Continued Pretraining of Language Models, Parmar et al, 2024, https://arxiv.org/abs/2407.07263
>
> [2] Continual Pre-Training of Large Language Models: How to (re)warm your model? Gupta et al, 2023 https://arxiv.org/pdf/2308.04014

---

### Author Response · Authors · 2025-11-19
**Common Response to Reviewers and Updated Manuscript: Addition Quality Assessment (Section 3.3)**

We thank all the reviewers for their time and valuable feedback.

We are delighted that reviewers found **the contribution to open science and open source of our work commendable** (reviewer upxT), and recognized our proposed pipeline as **robust and proper** (reviewer upxT),  describing it as **very sound** (reviewer "yQeP"), that **redesigns the text extraction pipeline for web data curation** (reviewer AQgW), effectively **avoiding information loss from naive HTML-to-text extraction** (reviewer yQeP).

We are pleased that reviewers considered our **domain-agnostic extraction pipeline** (reviewer yQeP) an **effective solution** (reviewer upxT), that **addresses the failure mode of previous web math extractors** (reviewer upxT). They also emphasized that it **reliably captures equations and maintaining code structure** (reviewers yQeP, AQgW), while **unifying MathJax/KaTeX/MathML into LaTeX** - a contribution that they found **very important but often under-estimated in previous works** (reviewer yQeP).

We are glad that reviewers found our delivered dataset to be **larger and higher quality than existing open-source alternatives** (reviewer upxT), **delivering better data quality than SOTA corpora such as MegaMath and FineMath** (reviewer AQgW), and that our **experiments clearly demonstrate the dataset’s quality** (reviewer yQeP).

Reviewers also described our **experimental results as strong** (reviewer upxT), **solid** (reviewer AQgW), and **showing very promising results** (reviewer yQeP). They highlighted our ablation on different refinement models being solid (reviewer AQgW) and noted that **our qualitative examples clearly demonstrate the superiority of our pipeline in preserving structure** (reviewer upxT). The **scale of our experiments was considered sufficient**, **showing good results across a range of benchmarks** (reviewer upxT).

We are further encouraged that reviewers described our paper as **well-written and structured** (reviewer AQgW).

***New Analysis Added:***

We have added a new section (Section 3.3) dedicated to a direct, quantitative assessment to rigorously compare the extraction fidelity of different datasets. We randomly sampled 100 documents from the shared subset of OWM, FineMath, MegaMath, and our corpus, evaluating four key aspects of content preservation:

1)  Math Preservation (0–3 or N/A): Correctness and completeness of mathematical expressions unified to LATEX.
2) Code Preservation (0–3 or N/A): Structural and semantic integrity, syntax, and functional behavior of preserved code blocks.
3) Faithfulness (0–3): Preservation of core scientific content integrity without omission or meaning alteration.
4) Readability (0–3): Overall clarity, organization, coherence, and textual flow of the output document.

We employed OpenAI gpt-5.1 as an automated judge to assess the conversion quality. The judge was provided with the original raw text (extracted via Lynx) and the converted document from each dataset, guided by a detailed scoring rubric and evaluation instructions (see Appendix A.6). We report the mean score for each dataset across samples and four dimensions.

The results show that our method preserves math and code substantially better than all baselines, including MegaMath (which also uses an LLM), due to our use of Lynx for stable rendering. While Faithfulness is slightly lower than FineMath and OWM—an expected and acceptable trade-off due to our LLM-based rewrite step that may compress or rewrite contextual details—it remains much higher than MegaMath. Crucially, our readability score is the highest among all datasets, decisively demonstrating that our approach successfully produces a corpus with superior textual coherence and clarity while retaining superior mathematical and code structure. We updated the paper and see Section 3.3.

Finally, we reaffirm our **commitment to open science** and will **release our full pipeline and dataset to the research community**.

---

### Author Response · Authors · 2025-12-02
**Response and Appreciation for Reviewer Feedback**

We thank all reviewers for their time and constructive feedback. We have carefully addressed all comments raised by the reviewers in the rebuttal and revised manuscript. We kindly ask Reviewer AQgW to consider updating their score in light of these revisions. We are also happy to clarify or address any remaining questions.

---

### Meta-Review · Area_Chair_6sbb · 2026-01-02

**Summary:**

This work introduces Nemotron-CC-Math pretraining dataset specifically for math LLM pretraining. Through a unified pipeline, which utilizes a Lynx render for various math formats and uses a targeted LLM to clean the data and converts the content to Latex format, the authors deliver two datasets Nemotron-CC-Math-3+ and 4+ with different sizes. Extensive experiments are conducted to show the great benefit from these datasets.

The reviewers mostly give pretty positive comments for the contribution of the datasets to the open-source committee. Therefore, the comments/concerns are mostly about some questions to further study the data.
Overall, this paper has no major weaknesses.

**Reviewer Concerns:**

The three reviewers all agree the great contribution of the dataset, the questions raised about the additional experiments, and also the "apple-to-apple" comparison are well addressed by the authors through further experiments during rebuttal.
Therefore, no big issues remained.

**Reviewer Scores:**

Reviewers have give quite high positive scores, I think Reviewer AQgW is also considerable to increase the score.

---

### Decision · Program_Chairs · 2026-01-26

Accept (Poster)